# Jagged1-Notch1-deployed tumor perivascular niche promotes breast cancer stem cell phenotype through Zeb1

Huimin Jiang[1,5], Chen Zhou[2,5], Zhen Zhang[1], Qiong Wang[1], Huimin Wei[1], Wen Shi[1], Jianjun Li[1], Zhaoyang Wang[1], Yang Ou[1], Wenhao Wang[1], Hang Wang[1], Quansheng Zhang[3], Wei Sun[1], Peiqing Sun [4] & Shuang Yang [1✉]

Zinc finger E-box binding homeobox 1 (Zeb1) has been demonstrated to participate in the acquisition of the properties of cancer stem cells (CSCs). However, it is largely unknown how signals from the tumor microenvironment (TME) contribute to aberrant Zeb1 expression. Here, we show that Zeb1 depletion suppresses stemness, colonization and the phenotypic plasticity of breast cancer. Moreover, we demonstrate that, with direct cell-cell contact, TME-derived endothelial cells provide the Notch ligand Jagged1 (Jag1) to neighboring breast CSCs, leading to Notch1-dependent upregulation of Zeb1. In turn, ectopic Zeb1 in tumor cells increases VEGFA production and reciprocally induces endothelial Jag1 in a paracrine manner. Depletion of Zeb1 disrupts this positive feedback loop in the tumor perivascular niche, which eventually lessens tumor initiation and progression in vivo and in vitro. In this work, we highlight that targeting the angiocrine Jag1-Notch1-Zeb1-VEGFA loop decreases breast cancer aggressiveness and thus enhances the efficacy of antiangiogenic therapy.

[1] Tianjin Key Laboratory of Tumor Microenvironment and Neurovascular Regulation, Medical College of Nankai University, 300071 Tianjin, China. [2] Xuanwu Hospital, Capital Medical University, 100053 Beijing, China. [3] Tianjin Key Laboratory of Organ Transplantation, Tianjin First Center Hospital, 300192 Tianjin, China. [4] Department of Cancer Biology, Wake Forest University School of Medicine, Winston-Salem, NC 27157, USA. [5] These authors contributed equally: Huimin Jiang, Chen Zhou. ✉email: yangshuang@nankai.edu.cn

Although cancer stem cells (CSCs) constitute only a small fraction of the total tumor cell population, they are thought to be the main drivers of tumorigenesis and cancer recurrence[1]. It has been demonstrated that specific changes in the tumor microenvironment (TME) may induce, through mutagenic or epigenetic alterations, a CSC expansion that potentiates tumor progression[2]. Notably, CSCs are not uniformly distributed within tumors but are rather enriched in perivascular niches, suggesting that CSCs critically interact with their TME[3–5]. However, how endothelial cells function to establish and maintain tumor-initiating-cell niches remains to be further elucidated. The development of CSC- and/or TME-targeting strategies that can eliminate the CSC population is critical for improving the clinical outcomes of patients with breast cancer.

Notch signaling, frequently upregulated in aggressive tumors[6,7], is initiated when transmembrane ligands on one cell bind to Notch receptors on an adjacent cell and cause γ-secretase-mediated proteolytic release of the Notch intracellular domain (NICD)[6]. NICD then translocates into the nucleus where it interacts with the transcriptional cofactor CBF1 and activates the transcription of target genes (e.g., *Hes1*) to modulate cell fate. Notch ligands from the Jagged (Jag) and Delta-like (Dll) families, frequently present on tumor endothelial cells[8,9], have been identified to promote Notch signaling in adjacent tumor cells[10–12]. For example, B-cell lymphoma cells produce FGF4 to upregulate Jag1 on neighboring endothelial cells, which in turn induces Notch2-Hey1 activation and enforces aggressive tumor phenotypes[12]. As such, via juxtacrine signaling between tumor cells and their surrounding endothelial cells, Notch activation can potentially generate a tumor perivascular microenvironment to orchestrate CSC behavior.

Zinc finger E-box binding homeobox 1 (Zeb1) is a transcription factor that influences developmental and homeostatic cell fates[13–15]. Importantly, aberrant Zeb1 expression has been shown to promote cancer progression in breast cancer and other cancer types[16–20]. Most studies suggest that ectopic ZEB1 is associated with a poor outcome in breast cancer patients based on its role in increasing tumorigenicity and stemness[21–23]. Mechanistically, Zeb1 acts as a transcriptional repressor of epithelial genes and cell polarity factors, thereby stimulating an undifferentiated and highly motile phenotype[19,24–26]. However, the microenvironmental cues that are responsible for the aberrant expression of Zeb1 in breast cancer remain largely unknown.

In this study, we provide evidence that TME-derived endothelial Jag1 elevates Zeb1 expression in neighboring breast CSCs in a Notch1-mediated juxtacrine manner. In turn, upregulation of Zeb1 in tumor cells increases VEGFA production and reciprocally induces endothelial Jag1 to create a positive feedback loop. Our data suggest that elucidating the microenvironmental signals (i.e., tumor perivascular niche) influencing aggressive breast cancer cells can provide effective treatment strategies.

## Results

**Zeb1 depletion reduces tumor initiation, grading, and distant metastasis.** To determine the role of Zeb1 in breast cancer initiation and progression, we crossed the floxed *Zeb1* allele homozygously into PyMT mice to generate PyMT;*Zeb1*[cKO] (*MMTV-Cre;PyMT;Zeb1*[fl/fl]) mice (Fig. 1a)[27]. Loss of Zeb1 expression in PyMT;*Zeb1*[cKO] tumor cells was confirmed by immunohistochemical staining and immunoblotting (Supplementary Fig. 1a, b). Moreover, an increase in E-cadherin and associated loss of Vimentin were demonstrated in PyMT;*Zeb1*[cKO] tumors. Similar to PyMT mice, all PyMT;*Zeb1*[cKO] mice developed breast cancer-like mammary tumors. However, depletion of *Zeb1* significantly delayed the onset and reduced the growth rate of

primary tumors (Supplementary Fig. 2a and Fig. 1b–d). Consistently, the number of Ki-67[+] proliferating tumor cells and blood vessel density markedly decreased, whereas the spontaneous apoptosis rate and deposition of extracellular matrix increased in PyMT;*Zeb1*[cKO] mice (Supplementary Fig. 1a). Notably, while PyMT tumors were often high grade and showed high intra- and intertumoral heterogeneity, the tumors in PyMT;*Zeb1*[cKO] mice displayed homogenous and mostly differentiated phenotypes (Supplementary Fig. 2b). Additionally, PyMT;*Zeb1*[cKO] mice showed a reduced capacity for lung colonization (Fig. 1e), as demonstrated by significant decreases in lung weight (Fig. 1f) and metastatic foci (Fig. 1g). *Zeb1* depletion also resulted in a remarkably prolonged survival rate in PyMT;*Zeb1*[cKO] mice (Fig. 1h). These observations collectively demonstrate that *Zeb1* depletion strongly reduces progression toward highly malignant, metastatic breast tumors.

**Zeb1 depletion reduces plasticity, stemness, and tumorigenic capacities.** In agreement with the strong heterogeneity of the PyMT primary tumors, corresponding tumor cells displayed highly variable mesenchymal-hybrid-epithelial phenotypes (Fig. 2a); however, all tumor cells derived from PyMT;*Zeb1*[cKO] mice were fixed in an epithelial state. The gene-set enrichment analysis (GSEA) further confirmed that loss of *Zeb1* expression shifted the cells towards an epithelial phenotype (Supplementary Fig. 3a and Fig. 2b). *Zeb1* depletion was also associated with reduced CSC signature and decreased metastatic competence (Fig. 2b). We further analyzed the expression of genes strongly associated with stemness and self-renewal phenotypes[28], demonstrating that all of the analyzed genes were expressed in PyMT cell lines but strongly downregulated upon *Zeb1* depletion (Supplementary Fig. 3b).

In addition, analysis of established breast CSC markers and properties[29] revealed strong decreases in the sphere-forming capacity (Fig. 2c), the CD44[+]CD24[−] breast CSC population (Fig. 2d) and the ALDH activity (Fig. 2e) in tumor cells from PyMT;*Zeb1*[cKO] mice compared to those from PyMT mice. Moreover, the PyMT;*Zeb1*[cKO] tumor cells were more sensitive to the chemotherapeutic agents epirubicin (EPI) and etoposide (ETOP) than the PyMT tumor cells (Fig. 2f, g). In in vivo extreme limiting dilution assays, we found that the tumor-initiating ability was significantly reduced in the PyMT;*Zeb1*[cKO] tumor cells compared with that in the PyMT cells (Fig. 2h). These data support the importance of Zeb1 in the acquisition of CSC properties by breast cancer cells, which leads to tumor initiation and progression.

**Notch1 activation contributes to increased Zeb1 expression.** To further investigate the tumor microenvironmental signals responsible for ectopic Zeb1 expression in breast cancer, we performed small-interfering RNA (siRNA) library screening using a human Zeb1 promoter construct and found that interference of the Notch1 receptor resulted in marked downregulation of Zeb1 transcription in MDA-MB-231 cells (Fig. 3a and Supplementary Table 1). Consistently, activation of Notch signaling using recombinant human Jag1 (rhJag1) upregulated Zeb1 expression at both the mRNA and protein levels (Fig. 3b, c). However, specific knockdown of Notch1, but not Notch2 and Notch3, strongly blocked this effect (Supplementary Fig. 4 and Fig. 3b, c). We also investigated Notch1-regulated expression of Zeb1 in SUM-159 cells and obtained the same results (Supplementary Fig. 5), revealing a predominant role for Notch1 in promoting Zeb1 expression[30].

Next, as shown in Fig. 3d, the wild-type −1915/+132 promoter of the human Zeb1 gene has two canonical Notch response

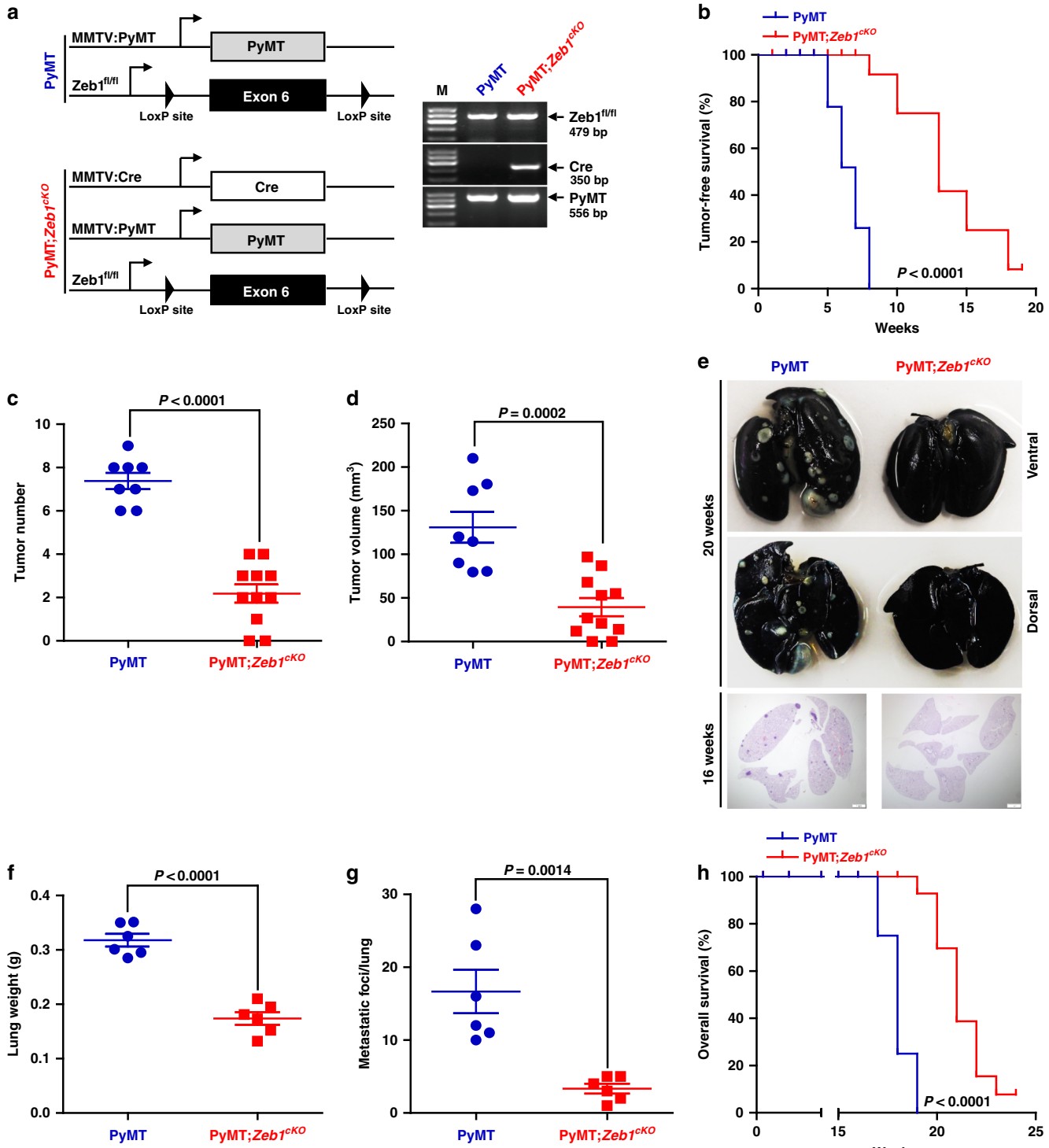

**Fig. 1 Zeb1 depletion reduces tumor development and metastasis in breast cancer. a** Scheme of the genetic mouse models for breast cancer. The color code (blue, PyMT; red, PyMT;Zeb1cKO) is used for all results. **b** Tumor-free survival ($n = 8$ PyMT, 11 PyMT;Zeb1cKO). **c** Tumor number ($n = 8$ PyMT, 11 PyMT;Zeb1cKO). **d** Tumor volume ($n = 8$ PyMT, 11 PyMT;Zeb1cKO). **e** Representative images of lung metastases at 16–20 weeks. Scale bars, 1 mm. **f** Lung weight ($n = 6$ PyMT, 6 PyMT;Zeb1cKO). **g** Metastatic lung foci ($n = 6$ PyMT, 6 PyMT;Zeb1cKO). **h** Overall survival ($n = 8$ PyMT, 12 PyMT;Zeb1cKO). Indicated $P$-values were calculated using log-rank test (Mantel–Cox) with multiple comparisons in **b**, **h** or two-tailed unpaired Student's $t$-test in **c**, **d**, **f**, **g**. Data are presented as mean ± SEM in **c**, **d**, **f**, **g**. Dots depict individual samples in **c**, **d**, **f**, **g**. Data are representative of at least five (**b**–**h**) independent experiments. Source data are provided as a Source Data file.

elements (NREs) at positions −761/−755 (GTGGGAA) and −384/ −378 (GTGGGTA), to which a complex comprising NICD and CBF1 may potentially bind[6]. The luciferase assay indicated that rhJag1 treatment increased the promoter activity of the Zeb1-p-2.0k reporter by approximately 1.7-fold relative to that of the control without rhJag1 addition (Fig. 3d). A series of truncated Zeb1 promoter-reporter constructs were then generated for analysis. The results showed that deletion or site-directed mutagenesis of either

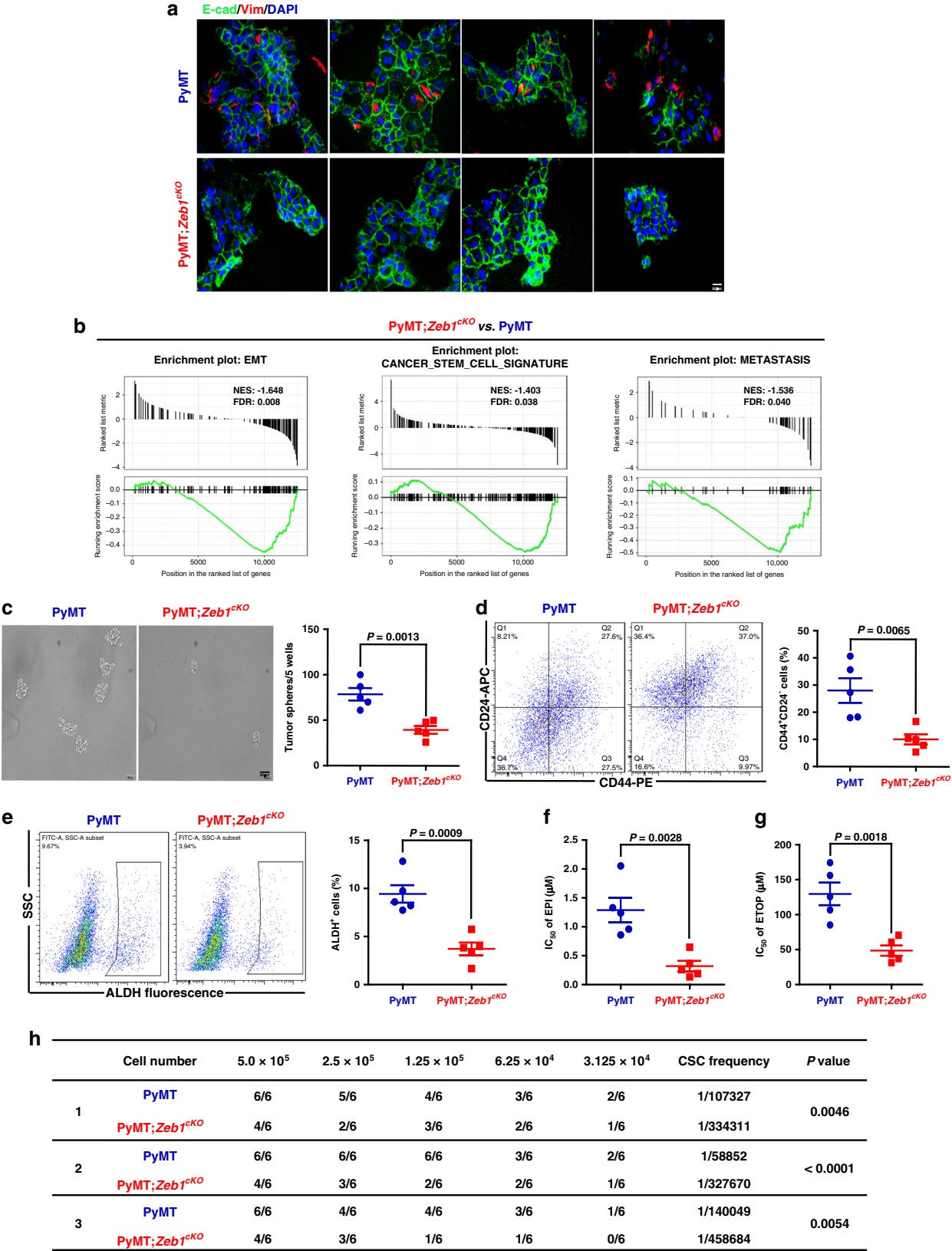

NRE did not affect rhJag1-induced transcriptional activation of the Zeb1 promoter, whereas simultaneous deletion or mutation of both NREs completely eliminated this effect (Fig. 3d, e). In contrast, treatment with the γ-secretase inhibitor DAPT showed the opposite effect of repressing the promoter activity of Zeb1 in a NRE-dependent manner (Supplementary Fig. 6). Mechanistically, the quantitative ChIP results demonstrated that NICD and CBF1 were recruited to both NREs in the Zeb1 promoter, and this recruitment was further increased by rhJag1 treatment (Supplementary Fig. 7 and Fig. 3f, g). These observations suggest

**Fig. 2 Depletion of Zeb1 affects the phenotypic variability of tumor cells. a** Immunofluorescence staining for E-cadherin and Vimentin showing variable expression in PyMT cell lines and homogeneous E-cadherin and lack of Vimentin expression in PyMT;Zeb1$^{cKO}$ cell lines. Scale bars, 50 µm. **b** GSEAs of transcriptome data from PyMT;Zeb1$^{cKO}$ vs. PyMT cells reveal enrichment of gene signatures associated with EMT, CSC phenotype, and reduced metastatic competence in PyMT;Zeb1$^{cKO}$ cell lines. NES, normalized enrichment score; FDR, false-discovery rate. **c** Tumor sphere formation analysis using single-cell suspension ($n = 5$ PyMT, 5 PyMT;Zeb1$^{cKO}$). Scale bars, 50 µm. **d, e** Flow cytometry analysis of **d** the CD44$^+$CD24$^-$ population and **e** ALDH activity ($n = 5$ PyMT, 5 PyMT;Zeb1$^{cKO}$). **f, g** IC$_{50}$ value following treatment with **f** EPI and **g** ETOP ($n = 5$ PyMT, 5 PyMT;Zeb1$^{cKO}$). **h** In vivo tumorigenicity assay with limited dilution ($n = 3$ PyMT, 3 PyMT;Zeb1$^{cKO}$). The estimated CSC frequency was analyzed using ELDA software. Indicated $P$-values were calculated using two-tailed unpaired Student's $t$-test in **c–g** or chi-square test with multiple comparisons in **h**. Data are presented as mean ± SEM in **c–g**. Dots depict individual samples in **c–g**. Data are representative of five (**a–g**) or three (**h**) independent experiments. Source data are provided as a Source Data file.

a predominant role for NREs in the regulation of Zeb1 transcription by Notch1 activation.

**Endothelial Jag1-activated Notch1 confers CSC phenotypes through Zeb1.** Given that the Notch ligands Jag1 and Dll4 are preferentially expressed by endothelial cells and play pivotal roles in establishing tumor-initiating-cell niches[8,9], we conducted a coculture experiment as shown in Fig. 4a. The results showed that coculture with HUVEC-mCherry increased Zeb1 expression in MDA-MB-231. However, depletion of Jag1, but not Dll4, markedly abrogated this effect (Supplementary Fig. 8 and Fig. 4a, b). On the other hand, upregulation of Zeb1 by cultured HUVEC-mCherry was specifically weakened in shNotch1/231 cells compared to cells expressing control shRNA or shRNAs for Notch2 or Notch3 (Fig. 4c, d). We also performed these experiments in a coculture system of SUM-159 with HUVEC-mCherry and obtained similar results (Supplementary Fig. 9), confirming that endothelial Jag1-induced juxtacrine activation of the Notch1 receptor leads to increased Zeb1 expression in breast cancer cells.

Importantly, our results demonstrated that coculture with HUVEC-mCherry led to increased stemness properties, including the CD44$^+$CD24$^-$ breast CSC population (Fig. 4e), the ALDH activity (Fig. 4f), the percentage of side population (Fig. 4g), and the expression of stemness-related genes (Fig. 4h), in shCtrl/231 cells, and these effects were strongly attenuated by Zeb1 depletion. In in vivo extreme limiting dilution assays, the tumorigenicity of shZeb1/231 cells was significantly reduced in response to coinjection with HUVECs compared with that of shCtrl/231 cells (Fig. 4i). These observations were further confirmed in the coculture system of shZeb1/159 with HUVECs (Supplementary Fig. 10), demonstrating that juxtacrine activation of Notch1 signaling by endothelial Jag1 induces Zeb1 expression in breast cancer cells, which in turn contributes to the establishment and maintenance of their CSC phenotypes.

**Zeb1 depletion subverts the tumor perivascular niche via regulating VEGFA.** We previously reported that increased expression of Zeb1 in breast cancer cells induces VEGFA production, thus promoting tumor angiogenesis and propagation[31]. Consistently, both the quantitative PCR and immunoblotting revealed decreased VEGFA expression in PyMT;Zeb1$^{cKO}$ tumor cells compared to that in PyMT cells (Fig. 5a, b). GSEA analysis also confirmed that loss of Zeb1 in PyMT;Zeb1$^{cKO}$ tumor cells was negatively enriched among genes associated with angiogenesis and VEGF-related signatures compared with that in PyMT tumor cells (Fig. 5c). Considering that VEGFA has been reported to enhance Jag1 expression in adventitial microvascular endothelial cells[32], this raised the possibility that depletion of Zeb1 in tumor cells might disrupt VEGFA-dependent paracrine actions to impair Jag1 activity in endothelial cells and subvert a malignant perivascular niche. Indeed, coculture with shCtrl/231 cells resulted in increased mRNA and protein levels of Jag1 in HUVEC-mCherry in a VEGFA-dependent manner, and these effects were

significantly attenuated by Zeb1 depletion (Fig. 5d, e). Additionally, we examined whether Zeb1 depletion in breast cancer cells would impair their interaction with endothelial cells. The results indicated that cells exhibiting loss of Zeb1 expression showed significantly reduced adhesion to a HUVEC monolayer and a decreased rate of tumor cell transmigration compared with shCtrl/231 cells (Fig. 5f, g). Consistently, HUVECs treated with conditioned media derived from shZeb1/231 cells had fewer F-actin stress fibers (Fig. 5h), which is a typical sign of an impaired endothelial barrier, than the cells treated with conditioned media from shCtrl/231 cells. Importantly, loss of Zeb1 expression in shZeb1/231 cells resulted in a weakened response to the VEGFA-neutralizing antibody Avastin in the coculture with HUVECs (Fig. 5f–h). We further confirmed these results in the coculture system of shZeb1/159 with HUVECs (Supplementary Fig. 11). Therefore, Zeb1 depletion in breast cancer cells impairs VEGFA paracrine signaling in adjacent endothelial cells to subvert a Jag1-mediated perivascular niche.

To determine whether this feedback loop drives breast tumorigenesis in vivo, CD31$^+$ tumor endothelial and EpCAM$^+$ breast cancer cells were isolated from PyMT;Zeb1$^{cKO}$ and PyMT mice. The results demonstrated that the fraction of CD31$^+$ endothelial cells was significantly reduced in PyMT;Zeb1$^{cKO}$ tumors (Fig. 6a), which was accompanied by decreased mRNA expression of Jag1 (Fig. 6b), compared to that in the PyMT tumors. The expression of NICD and Hes1 was also reduced in EpCAM$^+$ breast cancer cells from PyMT;Zeb1$^{cKO}$ mice compared with that from PyMT mice (Fig. 6c). Immunofluorescence staining for CD31, NICD and ALDH1 further confirmed strong decreases in microvessel density, Notch1 activity and CSC phenotype in PyMT;Zeb1$^{cKO}$ tumors (Fig. 6d, e). Notably, breast cancer cells expressing NICD and ALDH1 were predominantly located in the perivascular region close to endothelial cells in PyMT tumors; however, the perivascular colocalization of NICD and ALDH1 was significantly weakened by Zeb1 depletion. In line, the juxtacrine interaction between breast CSCs and adjacent endothelial cells were also confirmed to be Zeb1-dependent by immunofluorescence staining for CD31 and CD44 (Supplementary Fig. 12). Moreover, α-SMA (a pericyte marker) staining and dextran leakage assay demonstrated remarkably increased perivascular coverage and reduced dextran leakiness in PyMT;Zeb1$^{cKO}$ tumors (Fig. 6f, g), revealing a critical role for Zeb1 in the maintenance of tumor vessel normalization and vascular integrity[33]. Collectively, these observations suggest that endothelial cells within breast cancer might function as a CSC niche by providing Jag1 to the Notch1 receptor expressed in neighboring breast CSCs to elevate Zeb1 expression, which in turn reciprocally leads to the upregulation of endothelial Jag1 in a VEGFA-dependent paracrine fashion. Depletion of Zeb1 in breast cancer disrupts this positive feedback loop in the tumor perivascular niche, eventually decreasing CSC phenotypes and tumor angiogenesis.

**Zeb1 is positively correlated with Notch1 activity in human breast cancer.** To further strengthen the pathological correlation

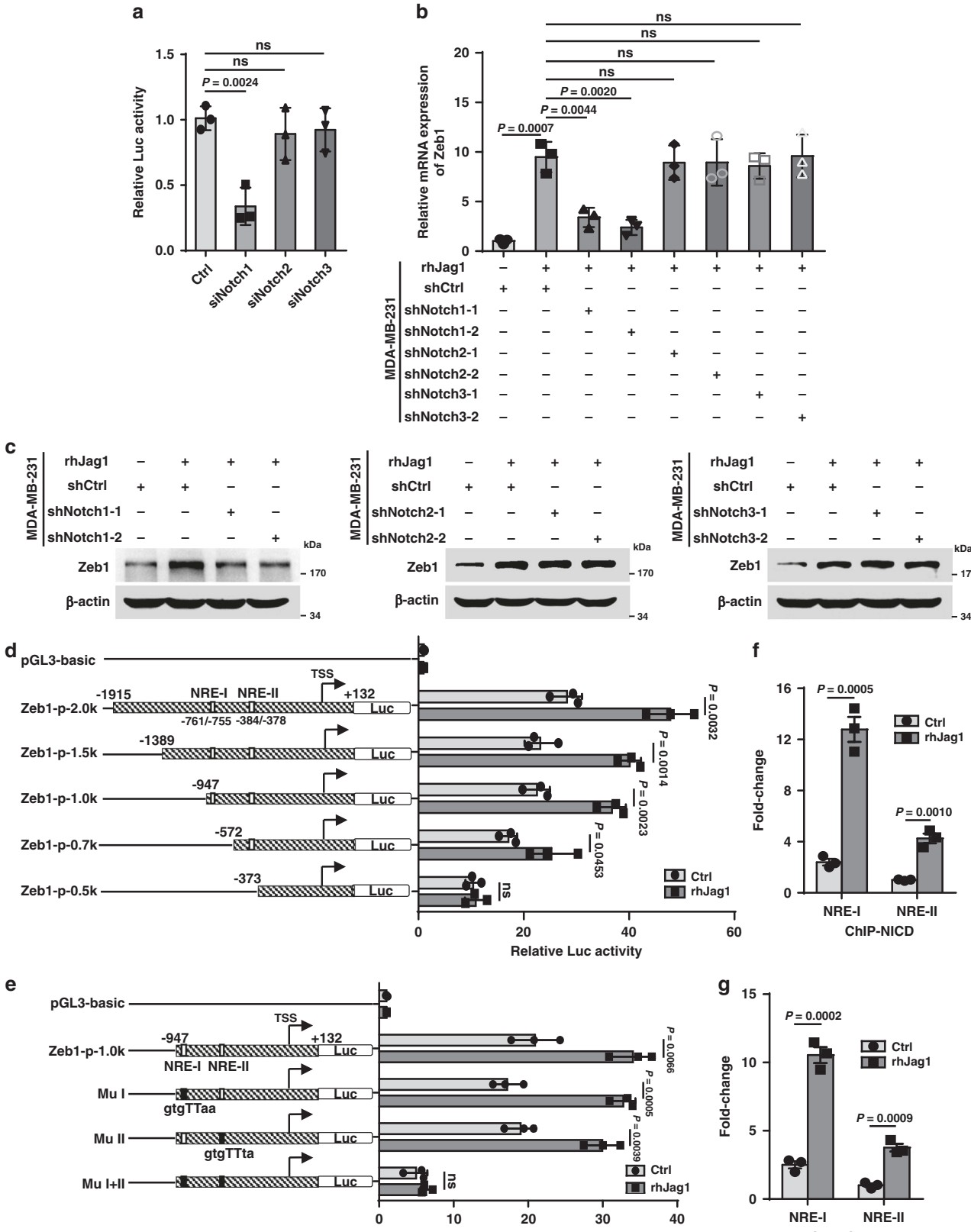

**Fig. 3 Notch1 activation upregulates Zeb1 expression. a** Luciferase assays for the wild-type promoter (−1915/+132) of Zeb1 in siNotch1-, siNotch2- or siNotch3-transfected MDA-MB-231 cells. **b** Relative mRNA levels of Zeb1 in shNotch1-, shNotch2- or shNotch3-transfected MDA-MB-231 cells after treatment with rhJag1. **c** Protein levels of Zeb1 in shNotch1-, shNotch2- or shNotch3-transfected MDA-MB-231 cells after treatment with rhJag1. **d** Luciferase assays for wild-type (−1915/+132) or truncated promoters of Zeb1 in MDA-MB-231 cells after treatment with rhJag1. **e** Luciferase assays for wild-type (−947/+132) or NRE-mutated promoters of Zeb1 in MDA-MB-231 cells after treatment with rhJag1. **f**, **g** ChIP assays for recruitment of **f** NICD and **g** CBF1 to the endogenous Zeb1 promoter in MDA-MB-231 cells after treatment with rhJag1. Indicated *P*-values were calculated using two-tailed unpaired Student's *t*-test. Data are presented as mean ± SEM in **a**, **b**, **d**–**g**. Data are representative of three (**a**–**g**) independent experiments. Source data are provided as a Source Data file.

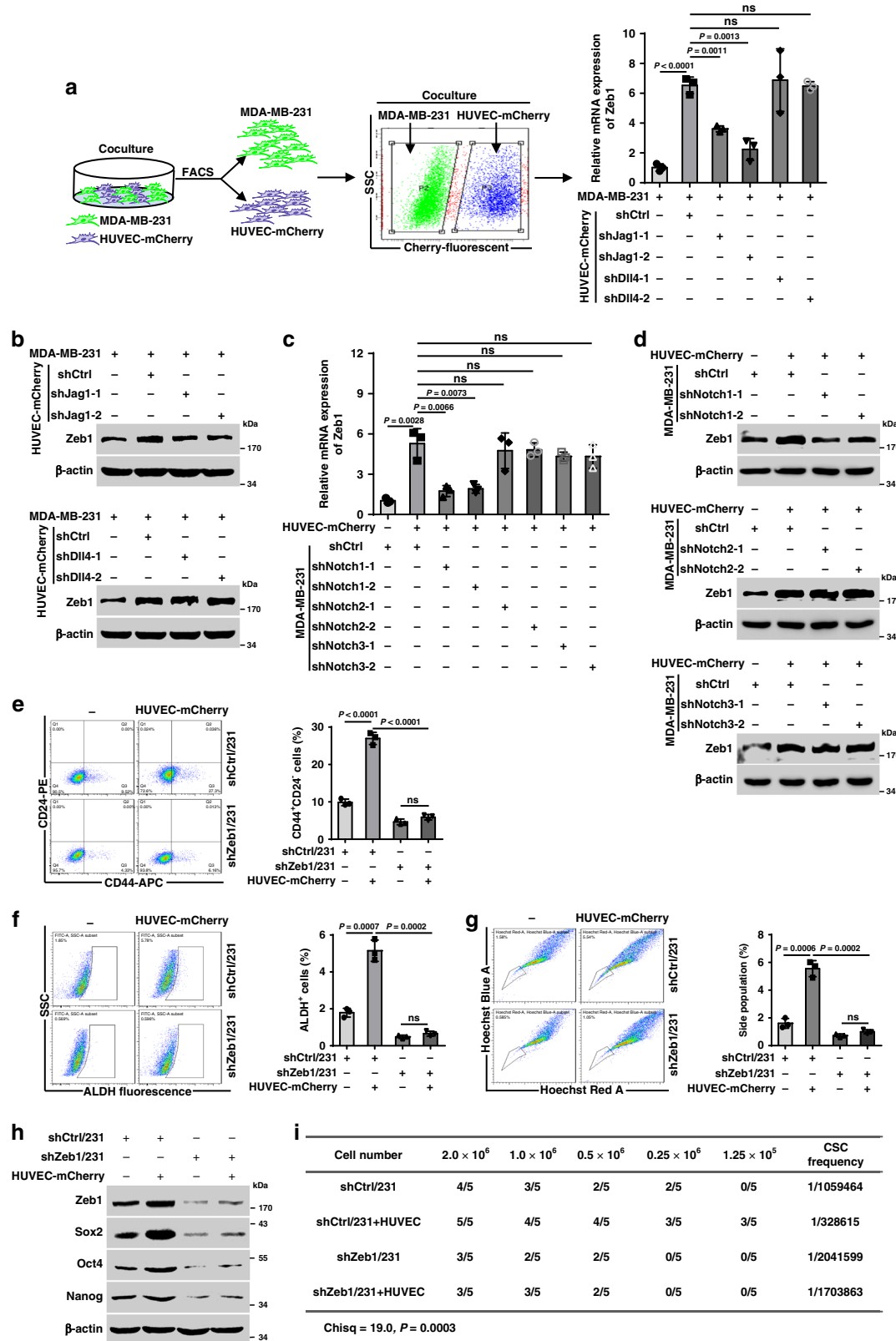

between Notch1-Zeb1 signaling and stemness properties in human breast cancer, we performed immunohistochemical staining for Zeb1, NICD and ALDH1 in 175 cases of primary breast carcinoma (Fig. 7a). The results indicated a strong positive correlation between the expression of Zeb1 and NICD (Fig. 7b). We also observed increased expression of Zeb1 and NICD in

tumors with high ALDH1 activity (Fig. 7c, d). Importantly, we demonstrated concomitantly high expression of Zeb1 and NICD in breast cancer patients with poorly differentiated high-grade tumors (Fig. 7e), highlighting that dysregulated Notch1-Zeb1 signaling is functionally linked to tumor stem cell traits in breast cancer.

**Fig. 4 Endothelial Jag1-induced activation of Notch1 contributes to CSC phenotypes through a Zeb1-dependent mechanism. a, b** Relative mRNA (**a**) and protein (**b**) levels of Zeb1 in MDA-MB-231 cells that were cocultured with shJag1- or shDll4-transfected HUVEC-mCherry. **c, d** Relative mRNA (**c**) and protein (**d**) levels of Zeb1 in shNotch1-, shNotch2- or shNotch3-transfected MDA-MB-231 cells that were cocultured with HUVEC-mCherry. **e–g** Flow cytometry analysis of **e** the CD44+CD24− population, **f** the ALDH activity, and **g** the side-population assay for shCtrl- or shZeb1-transfected MDA-MB-231 cells that were cocultured with HUVEC-mCherry. **h** Protein levels for Zeb1, Sox2, Oct4, and Nanog in shCtrl- or shZeb1-transfected MDA-MB-231 cells that were cocultured with HUVEC-mCherry. **i** In vivo tumorigenicity assay with limited dilution using shCtrl- or shZeb1-transfected MDA-MB-231 cells that were cocultured with HUVECs. The estimated CSC frequency was analyzed using ELDA software. Indicated P-values were calculated using two-tailed unpaired Student's *t*-test in **a**, **c**, **e–g**, or chi-square test with multiple comparisons in **i**. Data are presented as mean ± SEM in **a**, **c**, **e–g**. Data are representative of three (**a–h**) independent experiments. Source data are provided as a Source Data file.

Furthermore, our data revealed a very strong correlation between high expression of Zeb1 and NICD with the TNBC subtype (Fig. 7f, g). In contrast, low/medium expression of Zeb1 and NICD were positively correlated with the luminal subtype. Immunohistochemical staining for CD31 further confirmed a positive correlation between Zeb1/NICD expression and breast cancer angiogenesis (Supplementary Fig. 13), which is consistent with previous findings that high-grade tumors and TNBCs are often enriched with abundant CSC populations and are correlated with increased tumor angiogenesis[21,22,31,34].

**Targeting Zeb1-deployed perivascular niche reduces breast tumorigenesis.** Our study revealed a role for Jag1-Notch1-Zeb1-VEGFA-mediated angiocrine signaling in fostering tumor-initiating-cell niches in breast cancer. Therefore, combining Notch1 neutralizing antibody and Avastin treatment may suppress breast CSC phenotypes and further enhance the efficacy of antiangiogenic therapy. Thus, MDA-MB-231 and HUVECs were coinjected into the mammary fat pads of female BALB/c mice to allow the establishment of primary tumors, followed by treatment with IgG, anti-Notch1, Avastin, or both anti-Notch1 and Avastin (Fig. 8a). We noticed that, compared with the IgG control group, anti-Notch1 or Avastin treatment reduced the tumor volume (Fig. 8b) and weight (Fig. 8c) by 20–60%, while strong inhibition of tumor growth was achieved by anti-Notch1 and Avastin combinatorial treatment, with over 80% antitumor efficacy. The animal survival rate was also significantly prolonged in the combinational treatment group (Fig. 8d). Consistently, analysis of CD44+CD24− breast CSC population (Fig. 8e) and ALDH activity (Fig. 8f) displayed decreased breast CSC phenotypes in the tumors after single treatment with either anti-Notch1 or Avastin, while a synergistic inhibition was achieved by the combinational treatment. Immunoblotting for Zeb1, NICD, Hes1 and VEGFA also demonstrated strong downregulation of Zeb1 expression and decreases in Notch1 activity and VEGFA production in tumors from the combinational treatment group (Supplementary Fig. 14a). Moreover, the fraction of CD31+ endothelial cells was significantly reduced by the combinational treatment (Supplementary Fig. 14b), which was accompanied by decreased mRNA expression of Jag1 (Supplementary Fig. 14c). No significantly adverse effects, such as loss of animal body weight and toxic pathological changes in the heart, liver, spleen, lung, and kidney, were observed in anti-Notch1 and/or Avastin treatment group (Supplementary Fig. 15). Overall, these results imply that treating cancer patients with antiangiogenic therapy combined with anti-Notch1 might reduce cancer risk, which is achieved by inhibition of either CSC properties or tumor angiogenesis.

**Discussion**

A growing body of evidence has suggested that Zeb1 is a determinant of poor survival in human breast cancer. Therefore, the identification of microenvironmental signals that contribute to

ectopic Zeb1 expression may translate into improved antineoplastic therapies. Based on our findings, we propose that endothelial cells within the tumor function as a CSC niche by directly providing Notch1 ligands, such as Jag1, to the Notch1 receptors expressed in adjacent breast CSCs to elevate Zeb1 expression via a juxtacrine effect. Moreover, the properties of Zeb1-dependent perivascular niche are co-opted via activation of VEGFA production, which reciprocally induces endothelial Jag1 expression in proximity to neighboring CSCs. Thus, disruption of this microenvironmental juxtacrine/angiocrine loop at any level—such as Notch1 and VEGFA—severely diminishes the aggressiveness of breast cancer in vitro and in vivo (Fig. 8g).

In line with our results, various EMT transcription regulators (EMT-TFs) have been suggested to play important roles in tumor initiation and malignant progression[35]. However, we are beginning to understand the functional differences between Zeb1 and other EMT-TFs[17,22,36,37]. For instance, Zeb1 acts in strong contrast to Snail and Twist in the KPC mouse model of pancreatic cancer[17,36]. Depletion of *Zeb1* suppresses the stemness, colonization and phenotypic/metabolic plasticity of tumor cells[17]; however, depletion of *Snail* and *Twist* does not affect tumor differentiation, invasion or, importantly, metastasis in the same model[36]. In line, Zeb1 is much more powerful than Snail and Twist in promoting breast CSC-associated properties, such as radioresisitance, independently of its ability to induce the EMT program, highlighting that tumor onset is not necessarily related to adoption of a mesenchymal phenotype[22]. Of note, the expression of Twist is regulated by ectopic Zeb1 in radioresistant breast cancer cells[22], which is consistent with our RNA-sequencing result showing that the mRNA level of Twist was significantly reduced in tumor cells from PyMT;*Zeb1*cKO mice. Collectively, these findings point to functional differences in EMT-TFs, demonstrating that Zeb1 may exert different tumorigenic functions (e.g., stemness maintenance and EMT) that are not necessarily interrelated.

Notably, within tumors, high Zeb1 expression is restricted to a subpopulation of stem-like malignant cells at the invasive front[24,38], where cancer cells interact with their microenvironment. A few stimuli in TME have been identified to contribute to increased expression of Zeb1[30,39,40]. For instance, Chaffer et al.[39] reported that a dynamic model of non-CSC-to-CSC conversion in breast cancer is mediated through chromatin configuration at the Zeb1 promoter in response to TGF-β. In the present study, we provide further evidence that endothelial cells within the tumor function as a CSC niche by directly providing Jag1 to the Notch1 receptors expressed in breast CSCs to elevate Zeb1 expression. Our results also indicate strong heterogeneity in PyMT primary tumors with highly variable mesenchymal-hybrid-epithelial phenotypes. However, all tumor cells derived from PyMT;*Zeb1*cKO mice were fixed in an epithelial state, implying that Zeb1-enhanced plasticity of cancer cells represents ongoing transitions between an undifferentiated/(partial) mesenchymal and a differentiated/epithelial phenotype. This is consistent with the "migrating stem cell hypothesis" that highly metastatic tumor

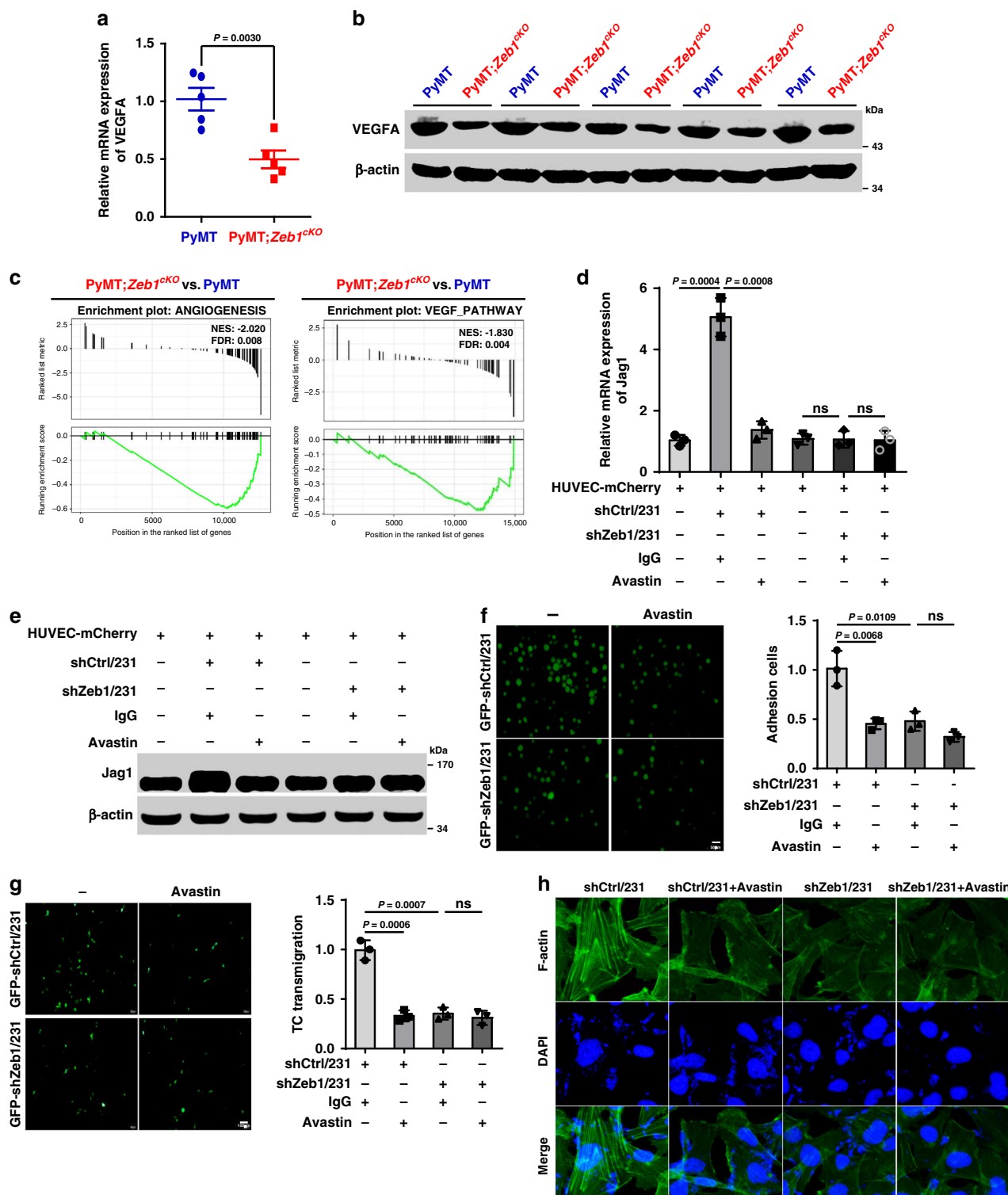

**Fig. 5 Depletion of Zeb1 impairs the tumor-endothelial cell interaction by downregulating VEGFA. a, b** Relative mRNA (**a**) and protein (**b**) levels of VEGFA ($n = 5$ PyMT, 5 PyMT;$Zeb1^{cKO}$). **c** GSEAs of transcriptome data from PyMT;$Zeb1^{cKO}$ vs. PyMT cells reveal enrichment of gene signatures associated with reduced angiogenesis and VEGF-related pathways in the PyMT;$Zeb1^{cKO}$ cell lines. **d, e** Relative mRNA (**d**) and protein (**e**) levels of Jag1 in HUVEC-mCherry that were cocultured with shCtrl- or shZeb1-transfected MDA-MB-231 cells in the presence of Avastin, a VEGFA-neutralizing antibody. **f** Tumor cell (shCtrl- or shZeb1-transfected MDA-MB-231) adhesion to HUVEC monolayers in the presence of Avastin. Scale bars, 50 μm. **g** Tumor cell (shCtrl- or shZeb1-transfected MDA-MB-231) transmigration through HUVEC monolayers seeded on transwell inserts in the presence of Avastin. Scale bars, 100 μm. **h** Immunofluorescence staining for phalloidin in F-actin filaments in HUVECs after treatment with conditioned media derived from shCtrl- or shZeb1-transfected MDA-MB-231 in the presence of Avastin. Scale bars, 20 μm. Indicated $P$-values were calculated using two-tailed unpaired Student's $t$-test. Data are presented as mean ± SEM in **a**, **d**, **f**, **g**. Dots depict individual samples in **a**. Data are representative of five (**a** and **b**) or three (**d–h**) independent experiments. Source data are provided as a Source Data file.

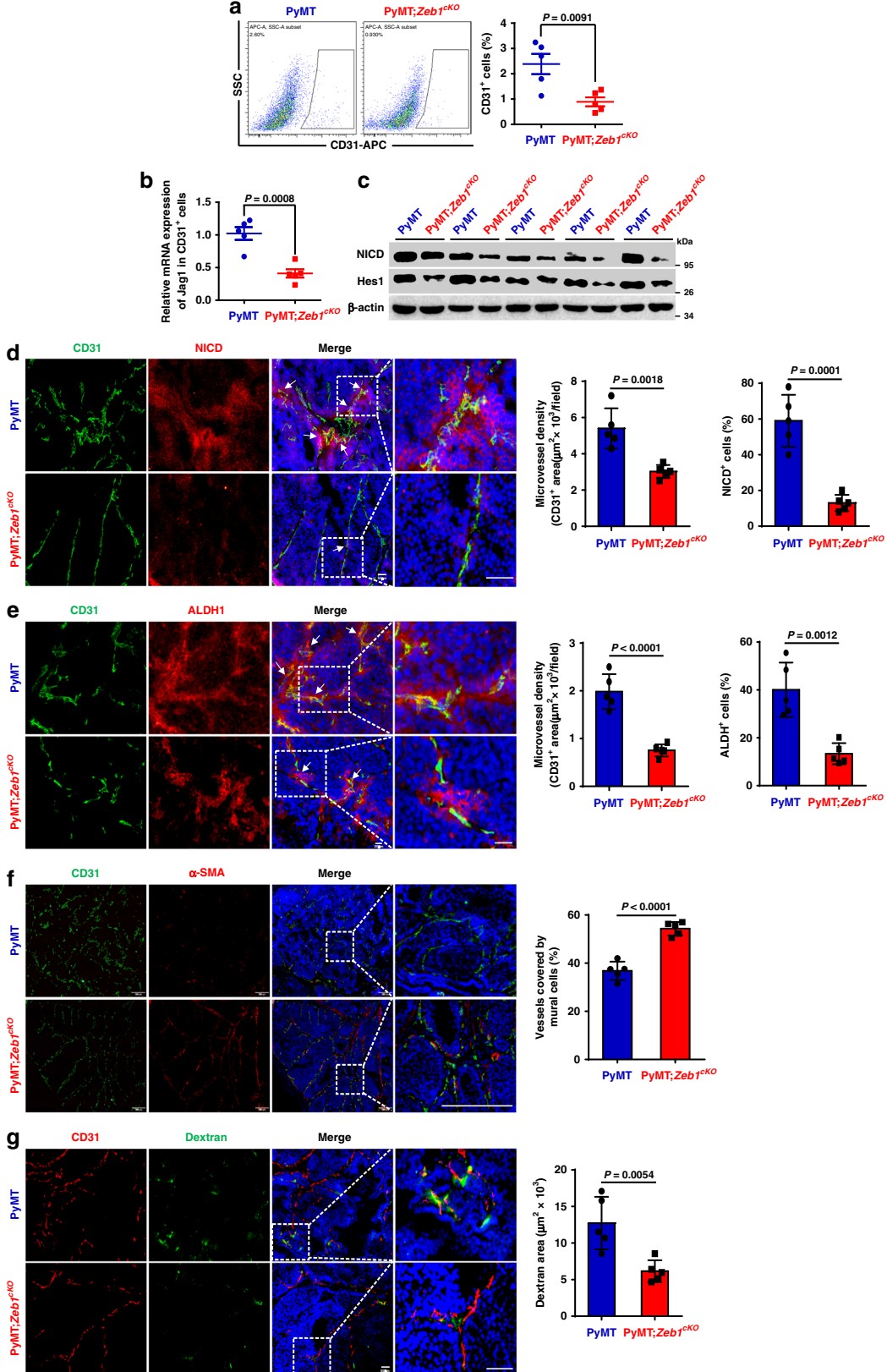

cells can transit reversibly between epithelial and mesenchymal states and acquire stem cell properties in response to extracellular signals[38,41,42]. Indeed, the Zeb1 gene itself exists in a poised, bivalent chromatin configuration, allowing a rapid switch between high expression in CSCs and low expression in non-CSCs in response to genetic and microenvironmental stimuli[39].

In addition, the perivascular niche-maintained CSC self-renewal can be achieved in multiple ways[4,42–44]. Our results provide further evidence that endothelial cells function as a stem cell niche for breast CSCs by directly providing the Notch1 ligand Jag1 to its receptors expressed in breast CSCs in a juxtacrine manner. Besides, a recent study has also shown that sphingosine-1-phosphate (S1P)

**Fig. 6 Depletion of Zeb1 subverts the tumor vascular niche. a** Flow cytometry analysis of the CD31$^+$ endothelial cell fraction ($n = 5$ PyMT, 5 PyMT; Zeb1$^{cKO}$). **b** Relative mRNA levels of Jag1 in CD31$^+$ endothelial cells ($n = 5$ PyMT, 5 PyMT;Zeb1$^{cKO}$). **c** Protein levels of NICD and Hes1 in EpCAM$^+$ tumor cells ($n = 5$ PyMT, 5 PyMT;Zeb1$^{cKO}$). **d** Immunofluorescence staining for NICD and CD31 in breast cancer tissue ($n = 5$ PyMT, 5 PyMT;Zeb1$^{cKO}$). The highlighted regions in the left panels are enlarged in the right panels. Scale bars, 50 μm. **e** Immunofluorescence staining for ALDH1 and CD31 in breast cancer tissue ($n = 5$ PyMT, 5 PyMT;Zeb1$^{cKO}$). Scale bars, 20 μm. **f** Immunofluorescence staining for CD31 and α-SMA in breast cancer tissue ($n = 5$ PyMT, 5 PyMT;Zeb1$^{cKO}$). Vessel normalization was determined with the CD31/α-SMA ratio. Scale bars, 200 μm. **g** Dextran leakage assay and immunofluorescence staining for CD31 in breast cancer tissue ($n = 5$ PyMT, 5 PyMT;Zeb1$^{cKO}$). Scale bars, 50 μm. Indicated $P$-values were calculated using two-tailed unpaired Student's $t$-test. Data are presented as mean ± SEM in **a**, **b**, **d**–**g**. Dots depict individual samples in **a**, **b**, **d**–**g**. Data are representative of five (**a**–**g**) independent experiments. Source data are provided as a Source Data file.

promotes CSC expansion via S1P receptor 3 and subsequent Notch1 activation in estrogen receptor (ER)-positive breast cancer cells[45], suggesting that activation of Notch1 signaling in breast CSCs might be mediated through both Notch ligand-dependent and -independent pathways. Moreover, it is important to note that, in addition to the canonical nuclear signaling, Notch1 also drives non-canonical, cytoplasmic and mitochondrial signals in TNBC stem cells[46–49], highlighting the generation of cell context-dependent diversity in the Notch1 signaling output. Nevertheless, the cause of the expression of Notch ligands in tumor perivascular niche is still unclear. Our results indicate that VEGFA produced by NICD/Zeb1-expressing breast CSCs acts as a short-range inter-cellular signal and upregulates Jag1 expression on neighboring endothelial cells. This is consistent with the previous studies showing that breast CSCs with elevated VEGFA expression contribute to angiogenesis or endothelial niche formation[50]. Taken together with our observation that combinational treatment with neutralizing antibodies targeting Notch1 and VEGFA synergistically reduces the development of breast tumors in xenograft models, our data indicate that disruption of the interaction between tumor cells and their perivascular niches would severely diminish the aggressiveness of breast cancer in vitro and in vivo.

In summary, our findings demonstrate that through a Zeb1-dependent mechanism, breast CSCs prime a maladaptive perivascular niche that reciprocally confers tumors with aggressive and lethal properties. Importantly, our study introduces promising therapeutic approaches to improve clinical outcomes for patients with aggressive breast cancer by ejecting CSCs from the protumorigenic perivascular niche to limit local tumor growth and aggressive progression.

## Methods

**Generation of conditional Zeb1$^{-/-}$ mice.** To generate the conditional Zeb1 knockout allele (Zeb1$^{fl/fl}$), exon 6 was flanked by loxP sites to remove sequences coding for large parts of the protein and to induce a premature translational stop. MMTV-PyMT mice were crossed with Zeb1$^{fl/fl}$ to generate Zeb1$^{-/-}$PyMT$^{+/-}$ mice (PyMT), which were then crossed with MMTV-Cre mice to generate Zeb1$^{-/-}$PyMT$^{+/-}$Cre$^{+/-}$mice (PyMT;Zeb1$^{cKO}$). PyMT and PyMT;Zeb1$^{cKO}$ off-spring were palpated weekly for tumor initiation. All mice were housed in a temperature-controlled room (22 ± 2 °C) with 40–60% humidity, with a light/dark cycle of 12 h/12 h. Mice were handled in accordance with protocols approved by the Animal Care and Use Committees of Medical College of Nankai University and Institute of Radiation Medicine of Chinese Academy of Medical Science.

**Genotyping.** PCR was performed using DNA from tail biopsies. All mice were genotyped to evaluate the *MMTV-PyMT*, *MMTV-Cre*, and *Zeb1* genes. The PCR primers for genotyping are listed in Supplementary Table 3.

**Primary cell lines.** Tumors were minced with a razor blade, rinsed three times with phosphate-buffered saline (PBS) and digested with collagenase I (Sigma) at 37 °C with agitation for 30 min in Dulbecco's Modified Eagle Medium (DMEM) with 2% fetal bovine serum (FBS) (BI). To establish tumor cell lines, $1 \times 10^7$ dissociated and filtered tumor cells were plated in a 10-cm dish in DMEM supplemented with 10% FBS, 1% sodium pyruvate and 1% Pen/Strep (Gibco). On the next day, dead cells were removed with a medium change, and the attached cells were used for subsequent experiments. Tumor cell lines were all derived from the high-grade carcinomas of 13-16-week-old females.

**Whole-mount mammary gland preparation.** The fourth mammary glands were surgically removed, stretched onto a glass slide and fixed in Carnoy's fixative (ethanol:chloroform:acetic acid, 6:3:1) overnight at room temperature (RT). The glands were then rehydrated in descending grades of alcohol (70%, 35% and 15%) for 10 min each step, rinsed with distilled water for 5 min, and stained overnight at RT in Carmine alum solution (Sigma). Further, mammary glands were dehydrated through a graded series of ethanol solutions (50%, 70%, 95%, and 100% alcohol) for 10 min each step, defatted in xylene overnight and stored in resinene (Sigma).

**Cell culture.** MDA-MB-231 and HUVEC cells were cultured in RPMI-1640 and SUM-159 cells were cultured in DMEM, supplemented with 10% FBS and 1% Pen/Strep. HEK293T cells were cultured in DMEM supplemented with 10% FBS, 1% sodium pyruvate, 1% NEAA and 2% glutamine.

**Plasmid construction.** The human complementary DNA (cDNA) fragment encoding the full-length Zeb1 sequence[51] was prepared by PCR and cloned into pLV-EF1-MCS-IRES-Bsd (Biosettia). The lentiviral-based vector pLV-H1-EF1α-puro (Biosettia) was used to express shRNAs in breast cancer cells and HUVECs. The human ZEB1 promoter (−1915/+132) sequences were obtained by PCR from human genomic DNA and cloned into the pGL3-basic vector (Promega). Mutagenesis of NRE-I and NRE-II in the human Zeb1 promoter was performed using a QuikChange® Lightning Site-Directed Mutagenesis kit (Stratagene). Primer sequences are listed in Supplementary Table 3.

**Generation of lentiviruses.** Lentiviruses were generated by transfecting sub-confluent HEK293T cells with lentiviral vectors and packaging plasmids by calcium phosphate transfection. Viral supernatants were collected 48 h after transfection, centrifuged at $75,000 \times g$ for 90 min, resuspended and filtered through 0.45-μm filters (Millipore).

**Mammosphere assay.** Single-cell suspension was suspended in sphere-culturing medium (Stemcell Technologies) at a density of 1000 cells/well and seeded into 24-well plates with ultra-low attachment surface (Corning) at 37 °C in a humidified atmosphere containing 5% $CO_2$. For evaluation, the mammospheres were counted and photographed under an inverted microscope (Olympus) after 2 weeks.

**Drug treatment.** Cells were seeded in a 96-well plate at a density of $3 \times 10^3$ cells/well, followed by treatment with serial concentrations of epirubicin (EPI) and etoposide (ETOP). Cell viability was measured at 48 h using the CCK-8 assay according to the manufacturer's protocols (Dojindo). Six parallel replicates were measured for each sample.

**Tumor cell implantation.** To measure tumor-initiating cell frequency, naturally arising tumors from 13- to 16-week-old females were collected. Tumor cells were dissociated into single-cell suspensions. Serial dilutions of sorted EpCAM$^+$ cells were suspended in 200 μl of a 1:1 PBS and Matrigel (Corning) mixture and injected into the fourth mammary fat pads of 4- to 6-week-old female BALB/c nude mice. In the coinjection experiments, serial concentrations of breast cancer cells were injected with or without HUVECs (1:3)[52]. The tumor-initiating cell frequency was determined using the ELDA webtool[53]. For the combined therapeutic experiments, MDA-MB-231 and HUVECs were coinjected. When the tumor volumes reached about 50 mm³, the mice were randomly divided into four groups, followed by intraperitoneal injection of IgG, anti-Notch1(10 mg/kg), Avastin(5 mg/kg), or both anti-Notch1 and Avastin twice a week. All animal studies were approved by the Institutional Animal Care and Use Committee of Medical college of Nankai University and Institute of Radiation Medicine of Chinese Academy of Medical Science.

**India ink assay.** Mice were euthanized and injected with 5 ml of India ink (15% v/v diluted in PBS; Hardy Diagnostics) into the trachea. The lungs were then removed and fixed in Fekete's solution (for 1 l, containing 880 ml of 70% ethanol, 80 ml of 37% formaldehyde, and 40 ml of glacial acetic acid) overnight at 4 °C. The lung metastasis nodules were counted manually.

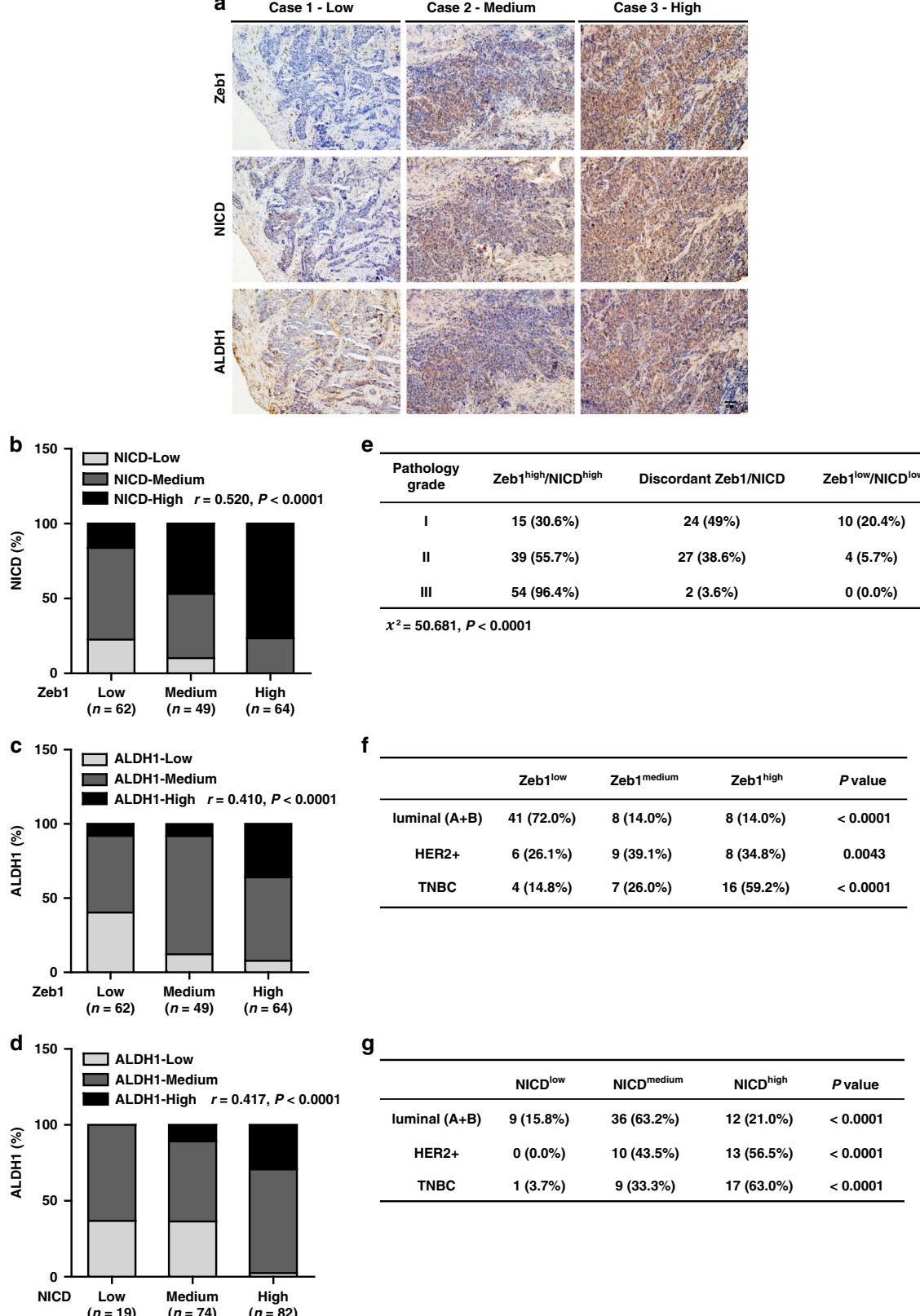

**Fig. 7 Zeb1 expression is positively correlated with enhanced Notch1 activity in aggressive human breast cancer. a** Representative images of immunohistochemical staining of Zeb1, NICD, and ALDH1 in serial sections of the same tumor ($n = 175$). Scale bars, 50 μm. **b** A positive correlation between the expression of Zeb1 and NICD in the 175 human breast cancer samples. **c** A positive correlation between the expression of Zeb1 and ALDH1 in breast cancer. **d** A positive correlation between the expression of NICD and ALDH1 in breast cancer. **e** Concomitantly high expression of Zeb1 and NICD in poorly differentiated high-grade breast cancer. **f, g** A strong correlation between high expression of Zeb1 (**f**) and NICD (**g**) in the TNBC subtype of breast cancer. Indicated *P*-values were calculated using Spearman's rank correction test in **b**–**d** or Kruskal–Wallis test in **e** or Kolmogorov–Smirnov test in **f, g**. Source data are provided as a Source Data file.

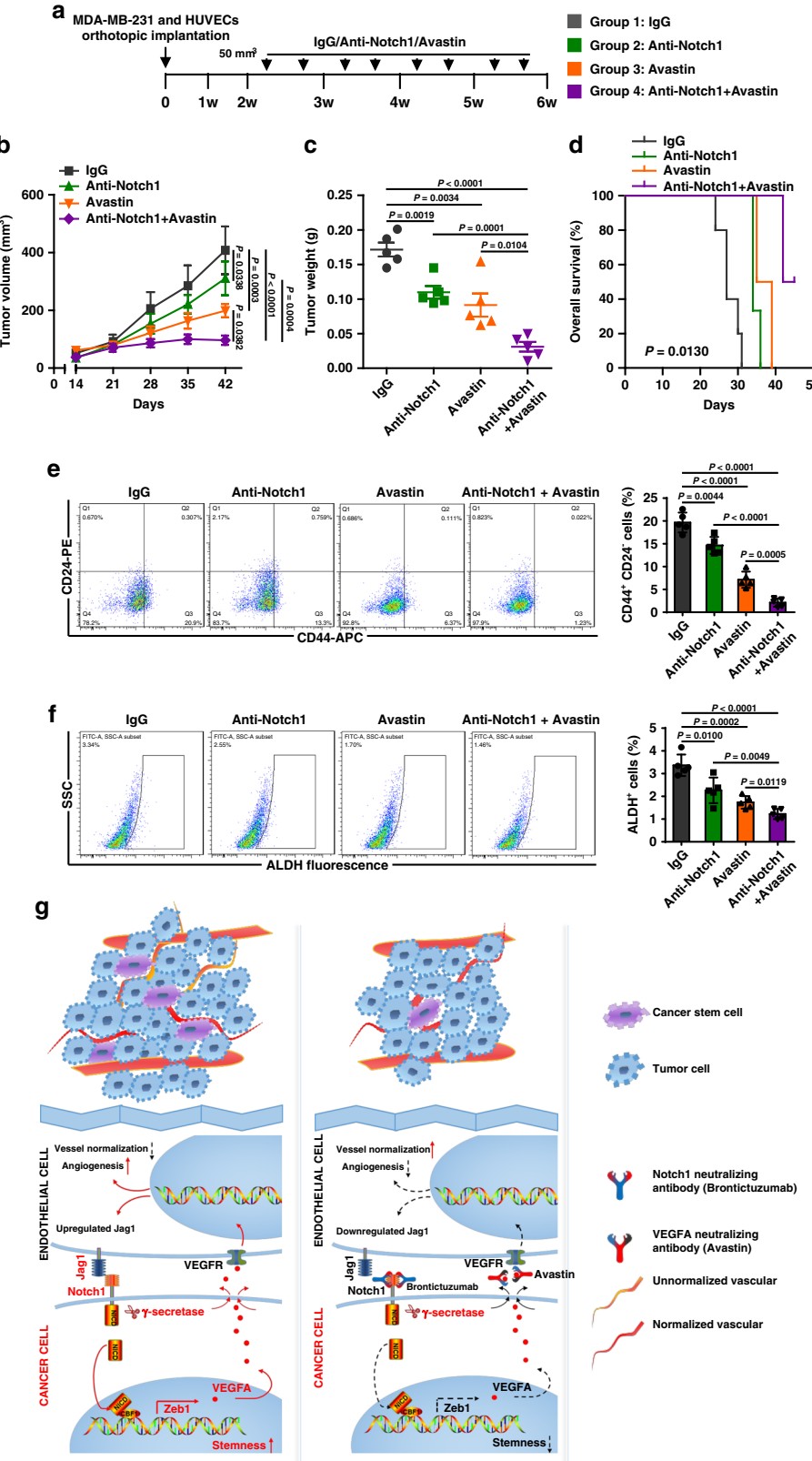

**RNA extraction and quantitative RT-PCR**. Total RNA from breast cancer cells and freshly sorted primary tumor cells was extracted using Trizol (Life Technologies). cDNA was synthesized using M-MLV Reverse Transcriptase (Takara). The specific products of Zeb1, VEGFA, Notch1-3, Jag1 and Dll4 were amplified by quantitative PCR using TransStart Green Q-PCR SuperMix Kit (TransGen). GAPDH was used as a normalization control. Primer sequences are listed in Supplementary Table 3.

**Immunoblotting assay**. Cells lysates were prepared in RIPA buffer with protease and phosphatase inhibitors. Protein concentrations was measured using BCA (Thermo) and equal amounts of proteins were loaded on a SDS-polyacrylamide electrophoresis (PAGE) gel. Transfer to polyvinylidene fluoride membranes (Bio-Rad) for further processing, membranes were next incubated overnight at 4 °C with primary antibodies followed by 1 h room temperature incubation with secondary antibodies. The appropriate antibodies were used as described in Supplementary

**Fig. 8 Combinatorial treatment of anti-Notch1 and Avastin synergistically reduces the incidence of breast tumorigenesis. a** Schematic showing the experimental approach used to examine whether disruption of the tumor perivascular niche by combinatorial treatment with anti-Notch1 and Avastin is essential for the inhibition of breast tumorigenesis. **b, c** Volume (**b**) and weight (**c**) of tumors from BALB/c mice coinjected with MDA-MB-231 and HUVECs following treatment with IgG, anti-Notch1, Avastin, or both anti-Notch1 and Avastin; $n = 5$. **d** Overall survival of mice with the indicated treatment; $n = 5$. **e, f** Flow cytometry analysis of (**e**) the $CD44^+CD24^-$ population and (**f**) ALDH activity of tumor cells from mice under the indicated treatment; $n = 5$. **g** Schematic of Zeb1-mediated remodeling of the breast cancer perivascular niche. Indicated $P$-values were calculated using LSD test with multiple comparisons in **b** or log-rank test (Mantel–Cox) with multiple comparisons in **d** or two-tailed unpaired Student's $t$-test in **c, e, f**. Data are presented as mean ± SEM in **b–f**. Source data are provided as a Source Data file.

Table 2. Labeled proteins were visualized by an ECL chemiluminescence kit (Millipore).

**siRNA library screening**. The Zeb1-promoter-Gluc plasmid was purchased from GeneCopoeia[TM]. An siRNA library of human growth factors and receptors (Sigma SI04100) was introduced into MDA-MB-231 cells, and luciferase activity was measured according to the instructions provided in the Secrete-Pair[TM] Dual Luminescence Assay Kit (GeneCopoeia[TM]).

**Luciferase assay**. Cells were transfected with wild-type or mutant human Zeb1 promoters, followed by treatment with rhJag1 (Sino Biological) or DAPT (Selleck) in 24-well plates. Lysates were prepared 36 h after transfection, and luciferase activity was measured using the Dual-Luciferase Reporter Assay System (Promega) according to the manufacturer's protocols. Luciferase activity was normalized to the values for Renilla luciferase.

**Chromatin immunoprecipitation (ChIP)**. ChIP assays were performed using an EZ-ChIP kit (Millipore) according to the manufacturer's instructions. Briefly, the cells were crosslinked with 1% formaldehyde for 10 min at RT, and the formaldehyde was then inactivated by the addition of 125 mM glycine. Chromatin extracts containing DNA fragments were immunoprecipitated using specific antibodies. The ChIP-enriched DNA was then uncrosslinked and subjected to quantitative PCR. The primers and antibodies used in these experiments are shown in Supplementary Tables 2 and 3.

**Immunoprecipitation assay**. Cell lysates were incubated with specific antibodies and Protein G agarose beads (Invitrogen) overnight at 4 °C, followed by three washes with a buffer containing 50 mM Tris (pH 7.5), 100 mM NaCl, 7.5 mM EGTA, and 0.1% Triton X-100. The pellet was then subjected to immunoblotting analysis. The antibodies used for immunoprecipitation are shown in Supplementary Table 2.

**Tissue samples**. A total of 175 breast cancer patient samples were obtained from the General Hospital of the People's Liberation Army (PLAGH, Beijing, China) along with pathological information. All patients had been histologically confirmed to have invasive ductal carcinoma of the breast and were recruited by the same department. This study was approved by the institutional ethics committees at PLAGH and Medical College of Nankai University. All patients provided informed consent according to the latest version of the Helsinki Declaration on human research ethics.

**Blood vessel leakiness analysis**. For the evaluation of tumor vessel leakiness, mice were given 1 mg of 70-kDa lysine fixable fluorescein labeled dextran (Thermo Fisher). After 10 min, the mouse was subjected to whole-animal perfusion. The tumors were carefully excised and cut into two halves along the maximum diameter. Both halves were fixed with 4% formaldehyde overnight at 4 °C, and then frozen. The FITC/CD31 double-positive vessels were identified as perfused vessels and counted, and the fluorescence intensity of the FITC-only positive area (extravasated FITC-dextran) was standardized for the perfused vessel counts.

**Immunohistochemical analysis**. Immunohistochemical analysis of paraffin-embedded sections was performed using the Envision Kit (Dako) following the manufacturer's protocols. Sections were boiled in retrieval solutions to expose antigens. Specific antibodies were applied to the sections. Slides were counter-stained with hematoxylin, dehydrated, and mounted. The H-score method was used to evaluate the percentage of positively stained cells and staining intensity. Immunostaining was independently evaluated by two pathologists. The antibodies used in the experiment are shown in Supplementary Table 2.

**Immunofluorescence microscopy**. Tissue sections or primary cells seeded on coverslips were permeabilized with 0.4% Triton X-100 for 15 min at RT, blocked with 5% goat serum for 1 h at RT and incubated with primary antibodies overnight at 4°C. Samples were then washed in TBST and incubated with appropriate Alexa Fluor 488- or Alexa Fluor 594-conjugated secondary antibodies (1:200, ZSGB-BIO)

for 1 h at RT. Nuclei were stained with DAPI (Sigma) for 2 min at RT. Images were acquired with a FV1000 confocal microscope (Olympus) and analyzed with ImageJ. The antibodies used in the experiment are shown in Supplementary Table 2.

**Coculture adhesion assay**. HUVECs were seeded into culture dishes and grown to an ~70% confluent monolayer after 18 h. Tumor cells were pelleted by centrifugation at $300 \times g$, resuspended in medium at $1 \times 10^6$ cells/ml, and laid on top of the cell monolayer. The nonadherent tumor cells were gently rinsed away after 1 h with two PBS washes, and the adherent cells were imaged based on GFP expression.

**Extravasation assay**. HUVECs were seeded into the upper chambers of Matrigel-coated transwell inserts and cultured in basal medium for 24 h to reach a monolayer. Breast cancer cells were then added on top of the HUVECs in medium without FBS. In the lower chambers, medium with 10% FBS was added. After 48 h, cells that migrated to the lower surface of the filter were counted by fluorescence microscopy. The number of extravasated cells was expressed as the mean number of cells per field over four fields in triplicate from at least three experiments.

**Flow cytometry and cell sorting**. For cell surface marker analysis, cells were resuspended in PBS containing 2% FBS and stained with fluorescent-conjugated antibodies against CD44, CD24, CD31, and EpCAM for 45 min at 4 °C. For detection of ALDH activity, the ALDEFLUOR kit (Stemcell Technologies) was used according to the manufacturer's instructions. For side population (SP) analysis, cells were resuspended in 37 °C PBS containing 2% FBS at a density of $1 \times 10^6$ cells/ml. Cells were incubated with 5 μg/ml Hoechst 33342 (Sigma) in the presence or absence of 50 μM verapamil (Sigma) for 1.5 h at 37 °C in darkness with intermittent shaking. The gating strategy is shown in Supplementary Fig. 16.

For cell sorting experiments, single-cell suspensions were incubated with appropriate antibodies and were subsequently analyzed with a FACS Aria instrument (BD). The antibodies used in the experiment are shown in Supplementary Table 2. All experiments using primary cells in vitro were performed minimally in quintuplicate.

**RNA-sequencing and data processing**. RNA sequencing was performed following the pipeline of BGI-tech (BGI). Briefly, total RNA was isolated using Trizol reagent (Invitrogen). After DNase I treatment, magnetic beads with Oligo (dT) were used to isolate mRNA. Then, mRNA was fragmented; purified; resolved with elution buffer for end reparation and single nucleotide A (adenine) addition; and finally connected with adapters. The Agilent 2100 Bioanalyzer and ABI StepOnePlus Real-Time PCR System were used for the quantification and quality control of the sample library. Primary sequencing data were produced by Illumina HiSeq[TM] 2000. After quality control, raw reads were filtered into clean reads and then processed by the tophat and cufflinks algorithms. Raw sequencing data (in fastq format) are available on NCBI_SRA (accession number: PRJNA511636). Genes with RPKM < 1 in both the PyMT and PyMT;$Zeb1^{cKO}$ samples were removed as nonexpressed genes. The enrichment analysis with KEGG, GO and transcription factor targets were performed by R (ClusterProfiler package) with default parameters[54]. GSEA analysis was performed by R (ClusterProfiler package) with selected parameters: nPerm = 1000, minGSSize = 10, maxGSSize = 500, $P$-valueCutoff = 0.05, and pAdjustMethod = "BH".

**Statistical analysis**. Statistical analyses were performed using GraphPad Prism 5 and SPSS 22 software. All the data are presented as the mean ± SEM and represent three or five independent experiments. Spearman's rank correlation test was used to analyze the correlation of gene expression in tissue samples. One-way analysis of variance (ANOVA) was used to compare means between treatment groups. Where appropriate, Student's $t$-test for unpaired observations was applied. A $P$-value < 0.05 was considered significant. The $r$-value test was used to evaluate the correlation analysis.

**Reporting summary**. Further information on research design is available in the Nature Research Reporting Summary linked to this article.

## Data availability
RNA-seq data that support the findings of this study have been deposited in NCBI_SRA with accession number PRJNA511636. All the other data supporting the findings of this

study are available within the article and its Supplementary Information files and from the corresponding author upon reasonable request. Data used to generate the figures are available in the Source Data file provided with this paper.

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

## Acknowledgements

This work is supported by grants from the National Natural Science Foundation of China (No. 81972454; No. 81670600), the Postdoctoral Research Foundation of Beijing (No. 2020-ZZ-019) and the Postdoctoral Science Foundation of China (No. 2019M660723).

## Author contributions

H.M.J., C.Z., P.Q.S., and S.Y. conceived and designed the study. H.M.J., C.Z., Z.Z., Q.W., H.M.W., W.Shi., H.W., W.Sun., and S.Y. developed methodology. H.M.J., C.Z., Z.Z., J.J.L., Z.Y.W., Y.O., W.H.W., H.W., Q.S.Z., W.Sun., and S.Y. analyzed and interpreted the data (for example, statistical analysis, biostatistics, computational analysis). H.M.J., C.Z., P.Q.S., and S.Y. wrote the paper.

## Competing interests

The authors declare no competing interests.
