## [Peer Review File · Nature Communications]

Reviewers' comments:

Reviewer #1 (Remarks to the Author); expert in Notch1

The manuscript convincingly demonstrates a paracrine loop between Jagged1, Notch1, Zeb1 and VEGF in the endothelial breast cancer stem cell niche. The observations presented are novel and translationally relevant. Experimental design is generally rigorous and the results are clearly presented. The manuscript would be further improved by addressing the following issues:

1. Lines 46-48: the activation of Notch receptors isn't always ligand-dependent. In fact, ligand-independent activation of Notch1 has been described in ER+ breast cancer stem cells. This should be discussed.
2. Lines 49-50: in addition to canonical nuclear signaling, Notch1 drives non-canonical, cytoplasmic and mitochondrial signals in triple-negative breast cancer stem cells. This should be discussed.
3. Line 81: "breast cancer" is not accurate when referring to mice. "breast cancer-like mammary tumors" is more accurate.
4. Legend to Figure 1: "pancreatic" cancer should be replaced with "mammary" cancer
5. The manuscript makes extensive use of siRNAs to Notch1, 2, 3 and Jagged1. The use of multiple siRNAs for each target is a strength. However, the extent of knockdown should be documented in supplemental figures. Some of the secondary effects on protein levels (e.g., Figs. 4b, 5e) are relatively modest, and may be explained by incomplete knockdown.
6. The in vivo experiments in Figure 8 used bevacizumab (Avastin) and bronctuzumab (anti-Notch1). From the figure, it appears that the antibodies were administered daily and simultaneously. Is this accurate? What were the doses of antibodies used, and how were they selected? Were the antibodies obtained from their manufacturers (Genentech and Oncomed) under MTAs?

Reviewer #2 (Remarks to the Author); expert in endothelial and stem cells

This report from Jiang H et al demonstrates that tumor vascular microenvironment provides instructive aberrant signals that promote Notch-dependent progression of pancreatic and breast cancer cells. The authors have used genetic deletion of Zeb1 to show that this transcription factor regulates the expression of Notch1 on the tumor cells. Notch1 activation in turn promotes the induction of VEGF-A, which through upregulation of Notch-ligand, Jag1 drives the tumor growth and metastasis. Importantly, the authors show that combined inhibition of VEGF-A via a neutralizing monoclonal antibody to VEGF-A (Avastin) and Notch signaling treatment significantly increase the survival of the tumor bearing mice.

Comments:

Uncovering the molecular pathways that modulate the corrupted cross-talk between tumor cells and vascular niche could lead to designing new strategies to target tumor invasiveness and metastasis. Currently, the majority of the clinical trials with anti-angiogenic therapies are afflicted with narrow bandwidth; primarily targeting only VEGF-A /VEGFR2 or Tie2 pathways, with marginal benefits. This is because studies to combine anti-angiogenic treatments are lagging behind.

The authors in this paper have attempted to address this unmet need. This is well executed and well controlled and mechanistic study that has meticulously unraveled the dysfunctional two way conversation between tumor endothelial cells and tumor cells. The finding that subverted tumor endothelial cells through upregulation of Notch ligands induce Zeb1 leading to enhanced aggressiveness and metastasis is novel and important finding. The data in Figure 8 showing that combination of Notch along with VEGF-A signaling through recruitment of Zeb1 leads to decrease in tumor growth and increase survival of the host is highly novel and could lead to more effective

therapies.

Before publication, the authors might consider addressing these issues:

1) In most figures dealing with the targeting of Zeb1 and anti-angiogenic approaches the authors have primarily used CD31 to identify the blood vessels. As CD31 is also expressed on the hematopoietic cells and often CD31+ monocytes or macrophages could be detected perivascularly, then quantification and assessment of the normalization of vessels just based on CD31 staining might not be accurate. Inclusion of staining with other endothelial cell-specific markers, such as VE-cadherin, or Claudin-5 or VEGFR2 might be more quantitative and accurate.

2) Besides VEGF and Notch signaling pathways are there other druggable pathways that are targeted by Zeb1? Has the authors considered to look at or assess published data of Zeb1 ChIP analyses purview to identify other pathways that are targeted by Zeb1?

3) In Figure 8 the authors conclude that targeting VEGF-A results in tumor vascular normalization. What type of criteria the authors have used to determine that indeed this normalization is a consequence of VEGF-A or Notch targeting? For example, is there less vascular leakiness? Is there more homogenous distribution of Claudin5? Etc.....More detailed histological analyses of the tumors in the data presented in Figure 8 or pancreatic tumors in figure 1 are necessary to determine the status of the tumor vessels during this combined therapeutic approaches.

4) Previous trials with targeting Dll4 have resulted in major overactive-angiogenic vascular toxicities. These toxicities have been further aggravated with combination of targeting other angiogenic pathways. Have the authors analyzed the status of the capillary and large vessels in the animals that were treated with the combination of Notch and VEGF-A signaling in Figure 8? Since the survival of the mice was improved these data indicate that indeed their combined approach might not have induced any severe toxicities. However, in depth probing of the organs of these mice treated with combination therapy might show subtle defects that in clinical trials could be a major bottleneck in implementing such therapies, especially if those therapies are given along with chemotherapy.

5) Could the authors show that the so called tumor stem cells are actually localized to the perivascular niche close to endothelial cells or randomly distributed through the tumors?

Minor comment:

As there are various isoforms and types of VEGF family (VEGF-A:121,165,181, 206) and (VEGF-B, VEG-C, VEGF-D, VEGF-E) then the authors should refer the VEGF being study properly as VEGF-A.

Reviewer #3 (Remarks to the Author); expert in breast cancer/microenvironment/mouse models/CSC

Jiang et al suggested that the tumor microenvironment activates Zeb1 in cancer stem cells to provoke tumor aggressiveness. As a mechanism, the Jag1-expressing endothelial cells activate Notch signaling in neighboring breast cancer stem cells, which eventually induces Zeb1 expression. In addition, enhanced Zeb1 in cancer stem cells increases VEGF production that in turn activates Jag1 expression in the environmental endothelial cells. Although they suggested autocrine Jag1-Notch1-Zeb1-VEGF feedback loop in breast cancer aggressiveness, the only novel finding would be that the Notch signaling can directly induce Zeb1 expression in this manuscript. However, they did not provide any solid experimental evidence that can support their conclusion at all. For example,

there is no evidence that (1) shRNAs worked, (2) rhJag1 was functional, (3) anti-Notch1icd antibody worked and was specific, (4) the co-culture system with HUVEC and MBA-MB-231 worked and was appropriate to evaluate their purpose, and (5) the co-transplantation experiment with HUVEC and MBA-MB-231 into mammary fat pad worked well. In addition to their experimental problems, I very wonder whether cancer stem cells contact to endothelial cells. In this case, do cancer stem cells experience hypoxia? Do cancer stem cells produce VEGF? Is there any evidence that Jag1 expression in endothelial cells is induced by VEGF that was produced by cancer stem cells? In immunohistochemical analysis of human breast cancer tissues, Notch1icd is expressed in the nucleus or cytoplasm of cancer stem cells? Are Notch1icd-expressing cells cancer stem cells? If so, I very wonder the frequency of cancer stem cells in the cancer tissues? Was anti-Notch1icd antibody specific? There are too many technical, experimental, and conceptual problems.

Reviewer's comments:

Reviewer #1 (Remarks to the Author); expert in Notch1

The manuscript convincingly demonstrates a paracrine loop between Jagged1, Notch1, Zeb1 and VEGF in the endothelial breast cancer stem cell niche. The observations presented are novel and translationally relevant. Experimental design is generally rigorous, and the results are clearly presented. The manuscript would be further improved by addressing the following issues:

1. Lines 46-48: the activation of Notch receptors isn't always ligand-dependent. In fact, ligand-independent activation of Notch1 has been described in ER+ breast cancer stem cells. This should be discussed.

Answer: Besides ligand-dependent mechanism, recent studies have also indicated a ligand-independent activation of Notch1 in human cancer progression. For example, Hirata *et al.* reported that sphingosine-1-phosphate (S1P) promotes cancer stem cell (CSC) expansion via S1P receptor 3 and subsequent Notch1 activation in ER α -positive MCF-7 breast cancer cells, providing new insights into the lipid-mediated regulation of CSCs via ligand-independent Notch1 activation [1]. Several studies have also demonstrated that the estrogen/ER signaling induces ligand-independent activation of Notch1 and Notch target genes, suggesting a model in which in ER α -positive breast epithelial cells Notch1 is a hormone-modulated signal that controls breast cancer progression [2-4]. Thus, combinations including an antiestrogen and a Notch1 inhibitor may be effective in ER α -positive disease. The above information has been added in the Discussion portion of the revised manuscript.

- [1] Hirata N, Yamada S, Shoda T, et al. Sphingosine-1-phosphate promotes expansion of cancer stem cells via S1PR3 by a ligand-independent Notch activation. *Nature communications*, 2014, 5: 4806.
- [2] Rizzo P, Miao H, D'Souza G, et al. Cross-talk between notch and the estrogen receptor in breast cancer suggests novel therapeutic approaches. *Cancer research*, 2008, 68(13): 5226-5235.
- [3] Pupo M, Pisano A, Abonante S, et al. GPER activates Notch signaling in breast cancer cells and cancer-associated fibroblasts (CAFs). *The international journal of biochemistry & cell biology*, 2014, 46: 56-67.
- [4] De Francesco E M, Maggiolini M, Musti A M. Crosstalk between Notch, HIF-1 α and GPER in breast cancer EMT. *International journal of molecular sciences*, 2018, 19(7): 2011.

2. Lines 49-50: in addition to canonical nuclear signaling, Notch1 drives non-canonical, cytoplasmic and mitochondrial signals in triple-negative breast cancer stem cells. This should be discussed.

Answer: Activation through NICD/CBF1 is referred to as canonical Notch1 signaling, and in addition there are non-canonical modes of Notch1 signaling, including CBF1-independent Notch1 signaling [5]. Although we still have a quite limited understanding of how non-canonical Notch1 signaling works, it appears to be important in breast cancer progression [6, 7]. For example, in breast cancer cell lines including triple-negative breast cancer (TNBC) cells, non-canonical Notch1 signaling is known to regulate IL-6 expression, and IL-6, in turn, acts on tumor cells to further increase their oncogenic potential. In this report, cytoplasmic NICD interaction with the NF- κ B pathway induces IL-6 expression, highlighting the generation of cell context-dependent diversity in the Notch1 signaling output [8]. Moreover, Hossain *et al.* reported that Notch1-activated signaling cascade regulates mitochondrial respiration and fermentation in TNBC stem cells, suggesting a model for AKT- and IKK α -dependent mitochondrial metabolism and NF- κ B activity in the cytoplasm, which is distinct from Notch1 canonical roles [9]. The above information has been added in the Discussion portion of the revised manuscript.

- [5] Andersen P, Uosaki H, Shenje L T, et al. Non-canonical Notch signaling: emerging role and mechanism. *Trends in cell biology*, 2012, 22(5): 257-265.
- [6] Ayaz F, Osborne B A. Non-canonical notch signaling in cancer and immunity. *Frontiers in oncology*, 2014, 4: 345.
- [7] Liu Z, Teng L, Bailey S, et al. Epithelial transformation by KLF4 requires Notch1 but not canonical Notch1 signaling. *Cancer biology & therapy*, 2009, 8(19): 1840-1851.
- [8] Jin S, Mutvei A P, Chivukula I V, et al. Non-canonical Notch signaling activates IL-6/JAK/STAT signaling in breast tumor cells and is controlled by p53 and IKK α /IKK β . *Oncogene*, 2013, 32(41): 4892.
- [9] Hossain F, Sorrentino C, Ucar D A, et al. Notch Signaling Regulates Mitochondrial Metabolism and NF- κ B activity in Triple-Negative Breast Cancer Cells via IKK α -dependent Non-Canonical Pathways. *Frontiers in oncology*, 2018, 8: 575.

3. Line 81: “breast cancer” is not accurate when referring to mice. “breast cancer-like mammary tumors” is more accurate.

Answer: In our study, we used MMTV-PyMT and MMTV-PyMT;*Zeb1*^{cko} animal models to study the role of *Zeb1* depletion in breast cancer initiation and progression.

According to the previous studies [10, 11], when referring to mice, “breast cancer-like mammary tumors” is used in the revised manuscript.

[10] Qiu T H, Chandramouli G V R, Hunter K W, et al. Global expression profiling identifies signatures of tumor virulence in MMTV-PyMT-transgenic mice: correlation to human disease. *Cancer research*, 2004, 64(17): 5973-5981.

[11] Christenson J L, Butterfield K T, Spoelstra N S, et al. MMTV-PyMT and derived Met-1 mouse mammary tumor cells as models for studying the role of the androgen receptor in triple-negative breast cancer progression. *Hormones and Cancer*, 2017, 8(2): 69-77.

4. Legend to Figure 1: “pancreatic” cancer should be replaced with “mammary” cancer.

Answer: This has been corrected in the revised manuscript.

5. The manuscript makes extensive use of siRNAs to Notch1, 2, 3 and Jagged1. The use of multiple siRNAs for each target is a strength. However, the extent of knockdown should be documented in supplemental figures. Some of the secondary effects on protein levels (e.g., Figs. 4b, 5e) are relatively modest, and may be explained by incomplete knockdown.

Answer: The knockdown efficiency of specific shRNAs for Notch1-3 in MDA-MB-231 and SUM-159 breast cancer cells were verified by quantitative PCR and immunoblotting (Figure I-1). The corresponding results (Figures 3c and S5h) have been repeated in these new cell lines and replaced in the revised manuscript.

Figure I-1: The knockdown efficiency of specific shRNAs for Notch1-3 in MDA-MB-231 and SUM-159 breast cancer cells. * $P < 0.05$, ** $P < 0.01$, *** $P < 0.001$ vs. the respective control by unpaired Student's t -test.

The knockdown efficiency of specific shRNAs for Jag1 and Dll4 in HUVEC-mCherry were also verified by quantitative PCR and immunoblotting (Figure I-2). The corresponding results (Figures 4b and S9b) have been repeated in these new cell lines and replaced in the revised manuscript.

Figure I-2: The knockdown efficiency of specific shRNAs for Jag1 and Dll4 in HUVEC-mCherry. ** $P < 0.01$, *** $P < 0.001$ vs. the respective control by unpaired Student's t -test.

6. The in vivo experiments in Figure 8 used bevacizumab (Avastin) and bronctuzumab (anti-Notch1). From the figure, it appears that the antibodies were administered daily and simultaneously. Is this accurate? What were the doses of antibodies used, and how were they selected? Were the antibodies obtained from their manufacturers (Genentech and Oncomed) under MTAs?

Answer: For the co-injection experiments, the antibodies were administered twice a week, alone or in combination. The humanized anti-human Notch1 monoclonal antibody (brontictuzumab, Creative Biolabs, Lot No. TAB-H11-1804) was used at a final concentration of 10mg/kg [12, 13]. The humanized anti-human VEGFA monoclonal antibody (VEGF-19-M, Alpha Diagnostic) was used at a final concentration of 5mg/kg [14, 15]. This information has been updated in the revised manuscript.

[12] Agnusdei V, Minuzzo S, Frasson C, et al. Therapeutic antibody targeting of Notch1 in T-acute lymphoblastic leukemia xenografts. *Leukemia*, 2014, 28(2): 278.

[13] Ferrarotto R, Mitani Y, Diao L, et al. Activating NOTCH1 mutations define a distinct subgroup of patients with adenoid cystic carcinoma who have poor prognosis, propensity to bone and liver metastasis, and potential responsiveness to Notch1 inhibitors. *Journal of Clinical Oncology*, 2017, 35(3): 352.

[14] Falk A T, Barriere J, Francois E, et al. Bevacizumab: a dose review. *Critical reviews in oncology/hematology*, 2015, 94(3): 311-322.

[15] Feng Q, Zhang C, Lum D, et al. A class of extracellular vesicles from breast cancer cells activates VEGF receptors and tumour angiogenesis. *Nature communications*, 2017, 8: 14450.

Reviewer #2 (Remarks to the Author); expert in endothelial and stem cells

This report from Jiang H *et al* demonstrates that tumor vascular microenvironment provides instructive aberrant signals that promote Notch-dependent progression of pancreatic and breast cancer cells. The authors have used genetic deletion of Zeb1 to show that this transcription factor regulates the expression of Notch1 on the tumor cells. Notch1 activation in turn promotes the induction of VEGF-A, which through upregulation of Notch-ligand, Jag1 drives the tumor growth and metastasis. Importantly, the authors show that combined inhibition of VEGF-A via a neutralizing monoclonal antibody to VEGF-A (Avastin) and Notch signaling treatment significantly increase the survival of the tumor bearing mice.

Comments:

Uncovering the molecular pathways that modulate the corrupted cross-talk between tumor cells and vascular niche could lead to designing new strategies to target tumor invasiveness and metastasis. Currently, the majority of the clinical trials with anti-angiogenic therapies are afflicted with narrow bandwidth; primarily targeting only VEGF-A/VEGFR2 or Tie2 pathways, with marginal benefits. This is because studies to combine anti-angiogenic treatments are lagging behind.

The authors in this paper have attempted to address this unmet need. This is well executed and well controlled and mechanistic study that has meticulously unraveled the dysfunctional two-way conversation between tumor endothelial cells and tumor cells. The finding that subverted tumor endothelial cells through upregulation of Notch ligands induce Zeb1 leading to enhanced aggressiveness and metastasis is novel and important finding. The data in Figure 8 showing that combination of Notch along with VEGF-A signaling through recruitment of Zeb1 leads to decrease in tumor growth and increase survival of the host is highly novel and could lead to more effective therapies.

Before publication, the authors might consider addressing these issues:

1) In most figures dealing with the targeting of Zeb1 and anti-angiogenic approaches the authors have primarily used CD31 to identify the blood vessels. As CD31 is also expressed on the hematopoietic cells and often CD31+ monocytes or macrophages could be detected perivascularly, then quantification and assessment of the normalization of vessels just based on CD31 staining might not be accurate. Inclusion of staining with other endothelial cell-specific markers, such as VE-cadherin, or Claudin-5 or VEGFR2 might be more quantitative and accurate.

Answer: To further confirm the vessel normalization in tumors from PyMT and

PyMT;*Zeb1*^{CKO} mice, we used anti-Claudin-5 antibody (Invitrogen, 35-2500) to perform immunofluorescence staining (Figure II-1). Our data revealed significantly improved vessel normalization in PyMT;*Zeb1*^{CKO} tumors, which was consistent with the results obtained using anti-CD31 antibody.

Figure II-1: Immunofluorescence staining for Claudin-5 and α -SMA in tumors from PyMT and PyMT;*Zeb1*^{CKO} mice. *** $P < 0.001$ vs. the respective control by unpaired Student's *t*-test. Scale bars, 50 μ m.

Moreover, anti-VE-cadherin antibody (BD, 555289) was also used to perform the experiments (Figure II-2); however, the result was not as specific as those of CD31 and Claudin-5.

Figure II-2: Immunofluorescence staining for VE-cadherin, CD31 and Claudin-5 in tumors from PyMT and PyMT;*Zeb1*^{CKO} mice. Scale bars, 50 μ m.

Considering that the staining for anti-CD31 was more specific in our results and has been verified as a marker for tumor endothelial cells in previous reports [16, 17], we kept this data in the revised manuscript.

[16]Tian L, Goldstein A, Wang H, et al. Mutual regulation of tumour vessel normalization and immunostimulatory reprogramming. *Nature*, 2017, 544(7649): 250.

[17]Lu J, Ye X, Fan F, et al. Endothelial cells promote the colorectal cancer stem cell phenotype through a soluble form of Jagged-1. *Cancer cell*, 2013, 23(2): 171-185.

2) Besides VEGF and Notch signaling pathways are there other druggable pathways that are targeted by Zeb1? Have the authors considered to look at or assess published data of Zeb1 ChIP analyses purview to identify other pathways that are targeted by Zeb1?

Answer: A growing body of evidence has suggested that Zeb1 promotes tumor initiation and progression and determines a worse survival in breast cancer, indicating a rationale for potential interventions to deplete Zeb1 in the treatment of advanced cancer [18, 19]. However, few studies have been identified to efficiently impair Zeb1 expression and to perform translational significance. In this present study, the GSEA analysis indicated that *Zeb1* depletion was associated with decreased estrogen pathway signature in PyMT;*Zeb1*^{CKO} cells compared with that in PyMT cells (Figure II-3a), which is consistent with our previous investigation showing that Zeb1 is a crucial determinant of resistance to antiestrogen therapies in breast cancer [20]. Of note, *Zeb1* depletion was also associated with reduced p38 MAPK and PI3K/Akt activity (Figure II-3b and c). Considering that several p38 MAPK (*e.g.*, Ralimetinib, Losmapimod and VX-702) and PI3K/Akt (*e.g.*, Miltefosine) inhibitors are in clinical and preclinical applications of anticancer treatment, these signaling pathways could also be the potential druggable pathways targeted by Zeb1 in breast cancer.

Figure II-3: GSEA analysis showing enrichment of gene signatures associated with decreased estrogen, p38 MAPK and PI3K/Akt signaling in PyMT;*Zeb1*^{CKO} cell lines.

In addition, we have demonstrated that Zeb1 plays an important role in breast cancer chemoresistance by increasing homologous recombination (HR)-mediated DNA damage repair [21]. The ChIP-seq analysis identified several transcriptional target genes of drug resistance bound by Zeb1, such as ataxia-telangiectasia mutation (ATM) kinase, *etc.* Various ATM inhibitors are being tested in anticancer treatment [22], which warrant investigation as candidate chemosensitizing agents for breast cancer with high levels of Zeb1. In addition, because depletion of Zeb1 chemosensitizes breast cancer cells *in vitro* and *in vivo*, we suggest that Zeb1-targeting agents may

have the potential to be used as tumor chemosensitizers.

Thus, we are working on another project to elucidate the regulation of Zeb1 expression at the post-translational level in breast cancer (unpublished data). Briefly, we found that specific inhibition of CDK4/6 activity results in decreased protein stability of Zeb1, leading to blockage of tumor progression *in vitro* and *in vivo* (Figure II-4). Mechanistically, specific phosphorylation of USP51, a deubiquitinating enzyme, by CDK4/6 is necessary to deubiquitinate and stabilize Zeb1. Considering that deubiquitinating enzymes (*e.g.*, USP51) are amenable to pharmacological inhibition by small molecule inhibitors, targeting USP51 to reduce Zeb1 stability, in combination with CDK4/6 inhibitors, will represent a new therapeutic strategy to deplete Zeb1 protein and overcome metastasis and therapy resistance in breast cancer.

Figure II-4: Breast cancer metastasis regulated by CDK4/6-USP51-Zeb1 axis through a deubiquitination-dependent mechanism.

- [18]Caramel J, Ligier M, Puisieux A. Pleiotropic roles for ZEB1 in cancer. *Cancer research*, 2018, 78(1): 30-35.
- [19]Zhang P, Sun Y, Ma L. ZEB1: at the crossroads of epithelial-mesenchymal transition, metastasis and therapy resistance. *Cell cycle*, 2015, 14(4): 481-487.
- [20]Zhang J, Zhou C, Jiang H, et al. ZEB1 induces ER- α promoter hypermethylation and confers antiestrogen resistance in breast cancer. *Cell death & disease*, 2017, 8(4): e2732.
- [21]Zhang X, Zhang Z, Zhang Q, et al. ZEB1 confers chemotherapeutic resistance to breast cancer by activating ATM. *Cell death & disease*, 2018, 9(2): 57.
- [22]Weber A M, Ryan A J. ATM and ATR as therapeutic targets in cancer. *Pharmacology & therapeutics*, 2015, 149: 124-138.

3) In Figure 8 the authors conclude that targeting VEGF-A results in tumor vascular normalization. What type of criteria the authors have used to determine that indeed this normalization is a consequence of VEGF-A or Notch targeting? For example, is there less vascular leakiness? Is there more homogenous distribution of Claudin5? Etc.....More detailed histological analyses of the tumors in the data presented in Figure 8 or pancreatic tumors in figure 1 are necessary to determine the status of the tumor vessels during these combined therapeutic approaches.

Answer: The alterations of vascular normalization and leakiness were both examined in tumors from PyMT and PyMT;*Zeb1*^{CKO} mice. As shown in Figure II-5, the results confirmed significantly increased perivascular coverage but decreased dextran leakage in tumors from PyMT;*Zeb1*^{CKO} mice compared with those from PyMT mice. We also observed a more homogenous distribution of CD31 in tumor tissues from PyMT;*Zeb1*^{CKO} mice, demonstrating that depletion of Zeb1 might improve vessel normalization and maturation in breast cancer. In line with this, a previous report has indicated that inhibition of cyclooxygenase-2 (COX-2) activity modulates VEGF signaling and thus increases vessel normalization, at least partially, via a Zeb1-dependent mechanism [23], revealing a critical role for Zeb1 in the maintenance of tumor vascular integrity.

Figure II-5: Vessel normalization (a) and dextran leakage (b) assays in tumors from PyMT;*Zeb1*^{CKO} mice. ** $P < 0.01$, *** $P < 0.001$ vs. the respective control by unpaired Student's *t*-test. Scale bars, 200 μm (a); 50 μm (b).

Mechanistically, we proposed that the microenvironmental abundance of VEGFA

induced by ectopic Zeb1 promotes continual tumor angiogenesis, playing a predominant role in the production of an abnormal blood vessel network with increased vessel leakage [24]. In addition, angiogenesis within solid tumors is strongly driven by hypoxia and fueled by acidosis [25, 26]. This is consistent with our result of GSEA analysis that *Zeb1* depletion was associated with reduced HIF-1 α signature in PyMT;*Zeb1*^{CKO} cells compared with that in PyMT cells (Figure II-6a). The expression of Zeb1 and HIF-1 α were further confirmed to be downregulated in PyMT;*Zeb1*^{CKO} cells (Figure II-6b). Of note, the PyMT;*Zeb1*^{CKO} tumors demonstrated a decreased hypoxia by pimonidazole (PIMO) staining compared with the PyMT tumors (Figure II-6c). These observations, together with the previous investigation by van den Xu *et al.* [27], implied that ectopic Zeb1 confers a hypoxic tumor microenvironment with abundant angiogenic factors (*e.g.*, VEGFA), thus facilitating breast cancer angiogenesis and growth. Depletion of Zeb1 in cancer cells would result in reversal of these abnormalities and increase vascular normalization and maturation, as shown in our present study.

Figure II-6: Ectopic Zeb1 confers a hypoxic tumor microenvironment in breast cancer. (a) Enrichment of gene signature associated with HIF-1 α -related pathway in PyMT;*Zeb1*^{CKO} vs. PyMT cells by GSEA analysis. (b) Expression of Zeb1 and HIF-1 α in CD44⁺CD24⁻ breast CSCs or non-CSCs (n = 3 PyMT, 3 PyMT;*Zeb1*^{CKO}). (c) PIMO staining for hypoxic area in breast cancer tissue (n = 5 PyMT, 5 PyMT;*Zeb1*^{CKO}). **P* < 0.05, ***P* < 0.01, ****P* < 0.001 vs. the respective control by unpaired Student's *t*-test. Scale bars, 50 μm (c).

Here we have demonstrated a Notch1-Zeb1-VEGFA-deployed tumor perivascular niche that promotes breast CSC properties. Disruption of this positive feedback loop by targeting Notch1 and/or VEGFA in PyMT breast cancer *in vivo* would

downregulate Zeb1 expression and eventually improve vessel maturation to lessen tumor progression. In line with this, we observed significantly decreased dextran vessel leakiness (Figure II-7a) and increased vessel normalization (Figure II-7b) in PyMT tumors especially from the combinational treatment of anti-Notch1 and anti-VEGFA.

Figure II-7: Dextran leakage (a) and vessel normalization (b) assays in PyMT breast cancer by combinatorial treatment with anti-Notch1 and/or anti-VEGFA. * $P < 0.05$, ** $P < 0.01$, *** $P < 0.001$ vs. the respective control by unpaired Student's t -test. Scale bars, 50 μm .

[23] Kirane A, Toombs J E, Larsen J E, et al. Epithelial–mesenchymal transition

increases tumor sensitivity to COX-2 inhibition by apricoxib. *Carcinogenesis*, 2012, 33(9): 1639-1646.

[24] Liu L, Tong Q, Liu S, et al. ZEB1 upregulates VEGF expression and stimulates angiogenesis in breast cancer. *PloS one*, 2016, 11(2): e0148774.

[25] Viallard C, Larrivée B. Tumor angiogenesis and vascular normalization: alternative therapeutic targets. *Angiogenesis*, 2017, 20(4): 409-426.

[26] De Francesco E M, Maggiolini M, Musti A M. Crosstalk between Notch, HIF-1 α and GPER in breast cancer EMT. *International journal of molecular sciences*, 2018, 19(7): 2011.

[27] Xu W Y, Hu Q S, Qin Y, et al. Zinc finger E-box-binding homeobox 1 mediates aerobic glycolysis via suppression of sirtuin 3 in pancreatic cancer. *World journal of gastroenterology*, 2018, 24(43): 4893.

4) Previous trials with targeting Dll4 have resulted in major overactive-angiogenic vascular toxicities. These toxicities have been further aggravated with combination of targeting other angiogenic pathways. Have the authors analyzed the status of the capillary and large vessels in the animals that were treated with the combination of Notch and VEGF-A signaling in Figure 8? Since the survival of the mice was improved these data indicate that indeed their combined approach might not have induced any severe toxicities. However, in depth probing of the organs of these mice treated with combination therapy might show subtle defects that in clinical trials could be a major bottleneck in implementing such therapies, especially if those therapies are given along with chemotherapy.

Answer: For the co-injection experiment *in vivo*, no significantly adverse effects in the gross measures, such as loss of animal body weight, skin ulcerations and toxic death, were observed in anti-Notch1 and/or Avastin treatment group. Moreover, toxic pathological changes in the heart, liver, spleen, lung, kidney and ventral aorta were not detected by microscopic examination (Figure II-8).

Figure II-8: Examination of the body weight (a) and H&E staining for the heart, liver, spleen, lung, kidney and ventral aorta (b) in mice treated with anti-Notch1 and/or Avastin. Scale bars, 50 μ m.

5) Could the authors show that the so called tumor stem cells are actually localized to the perivascular niche close to endothelial cells or randomly distributed through the tumors?

Answer: To further demonstrate that breast CSCs are localized to the perivascular niche close to endothelial cells, we performed immunofluorescent staining in tumors from PyMT and PyMT;*Zeb1*^{CKO} mice using anti-CD31, anti-ALDH1 and anti-NICD antibodies. As shown in Figure II-9a and b, we observed that breast cancer cells expressing ALDH1 and NICD were predominantly colocalized with CD31-positive endothelial cells in tumors from PyMT mice, showing that breast CSCs with elevated Notch1 activity are largely abundant in the adjacent perivascular niches. However, this perivascular location of ALDH1 and NICD was significantly attenuated by *Zeb1* depletion in PyMT;*Zeb1*^{CKO} tumors. Further, our results revealed strongly decreased microvessel density, reduced CSC phenotype and Notch1 activity in PyMT;*Zeb1*^{CKO} tumors, highlighting that depletion of *Zeb1* contributes to the impairment of tumor

perivascular niche and inhibition in CSC properties.

Figure II-9: Immunofluorescence staining for CD31, ALDH1 and NICD in tumors from PyMT and PyMT;Zeb1^{CKO} mice (n = 5 PyMT, 5 PyMT;Zeb1^{CKO}). **P < 0.01, ***P < 0.001 vs. the respective control by unpaired Student's *t*-test. Scale bars, 20μm (a); 50μm (b).

In line, the immunofluorescence staining for CD31 and CD44 confirmed the colocalization of CD44⁺ breast CSCs in the perivascular region close to endothelial cells in PyMT tumors, whereas this contact between breast CSCs and endothelial cells was significantly disrupted in PyMT;Zeb1^{CKO} tumors (Figure II-10).

Figure II-10: Immunofluorescence staining for CD31 and CD44 in tumors from PyMT and PyMT;Zeb1^{CKO} mice (n = 5 PyMT, 5 PyMT;Zeb1^{CKO}). *P < 0.05, ***P < 0.001 vs. the respective control by unpaired Student's *t*-test. Scale bars, 20μm.

For *in vitro* assay using the co-culture system, we performed the immunofluorescence staining for mCherry, CD44 and CD24 using the co-culture of MDA-MB-231 with HUVEC-mCherry *in vitro*. The analysis of confocal microscopy demonstrated that the direct interaction with HUVEC-mCherry induced an increased CD44⁺CD24⁻ breast CSC phenotype in contact MDA-MB-231 but not in free MDA-MB-231 (Figure II-11). Taken together with our notions that endothelial Jag1-induced juxtacrine activation of Notch1 conferred breast CSC phenotypes in a Zeb1-dependent manner,

we proposed that the abundance of breast CSCs with dysregulated Notch1-Zeb1 activity are predominantly resided in the perivascular niches but are not randomly distributed through the tumors.

Figure II-11: Contact with HUVEC-mCherry promotes CD44⁺CD24⁻ breast CSC property in MDA-MB-231 cells in the co-culture system. *** $P < 0.001$ vs. the respective control by unpaired Student's t -test. Scale bars, 20µm.

These observations are consistent with several previous studies showing that the perivascular niches are localized to adjacent tumor-initiating cells which could directly interact with each other in certain types of cancers [28-31]. For example, the vascular niche localized in the subventricular zone houses neural stem and progenitor cells that could initiate gliomas [28, 29]. Taken together, the remarkable proximity of the vasculature to CSCs in solid tumors lends credence to the concept that endothelial cells might be pathophysiologically reprogrammed to support tumorigenesis once organ-specific stem and progenitor cells undergo malignant transformation. Alternatively, the perivascular niche might be endowed with the ability to sustain both the putative tumor-initiating cells and the tumor cell populations that do not have the capacity to initiate tumors but maintain tumor mass.

[28] Calabrese C, Poppleton H, Kocak M, et al. A perivascular niche for brain tumor stem cells. *Cancer cell*, 2007, 11(1): 69-82.

- [29]Zhu T S, Costello M A, Talsma C E, et al. Endothelial cells create a stem cell niche in glioblastoma by providing NOTCH ligands that nurture self-renewal of cancer stem-like cells. *Cancer research*, 2011, 71(18): 6061-6072.
- [30]Butler J M, Kobayashi H, Rafii S. Instructive role of the vascular niche in promoting tumour growth and tissue repair by angiocrine factors. *Nature Reviews Cancer*, 2010, 10(2): 138.
- [31]Meurette O, Mehlen P. Notch signaling in the tumor microenvironment. *Cancer Cell*, 2018, 34(4): 536-548.

Minor comment:

As there are various isoforms and types of VEGF family (VEGF-A:121,165,181, 206) and (VEGF-B, VEG-C, VEGF-D, VEGF-E) then the authors should refer the VEGF being study properly as VEGF-A.

Answer: This has been corrected in the revised manuscript.

Reviewer #3 (Remarks to the Author); expert in breast cancer/microenvironment/mouse models/CSC

Jiang et al suggested that the tumor microenvironment activates Zeb1 in cancer stem cells to provoke tumor aggressiveness. As a mechanism, the Jag1-expressing endothelial cells activate Notch signaling in neighboring breast cancer stem cells, which eventually induces Zeb1 expression. In addition, enhanced Zeb1 in cancer stem cells increases VEGF production that in turn activates Jag1 expression in the environmental endothelial cells. Although they suggested autocrine Jag1-Notch1-Zeb1-VEGF feedback loop in breast cancer aggressiveness, the only novel finding would be that the Notch signaling can directly induce Zeb1 expression in this manuscript. However, they did not provide any solid experimental evidence that can support their conclusion at all. For example, there is no evidence that:

(1) shRNAs worked;

Answer: The knockdown efficiency of specific shRNAs for Notch1-3 in MDA-MB-231 and SUM-159 breast cancer cells were verified by quantitative PCR and immunoblotting (Figure III-1). The knockdown efficiency of specific shRNAs for Jag1 and Dll4 in HUVECs were verified by quantitative PCR and immunoblotting (Figure III-2).

Figure III-1: The knockdown efficiency of specific shRNAs for Notch1-3 in MDA-MB-231 and SUM-159 breast cancer cells. * $P < 0.05$, ** $P < 0.01$, *** $P < 0.001$ vs. the respective control by unpaired Student's t -test.

Figure III-2: The knockdown efficiency of specific shRNAs for Jag1 and DLL4 in HUVECs. ** $P < 0.01$, *** $P < 0.001$ vs. the respective control by unpaired Student's t -test.

(2) rhJag1 was functional;

Answer: To confirm the efficacy of rhJag1 to activate Notch1 signaling in breast cancer cells, we treated MDA-MB-231 and SUM-159 with the indicated concentrations of rhJag1. As shown in Figure III-3, the results demonstrated a dose-dependent upregulation of NICD (Notch1cd) and Hes1 (a downstream target gene of Notch1 signaling) expression, which is consistent with the previous studies that stimulation of Notch1 signaling by rhJag1 addition contributes to NICD upregulation and thus breast cancer initiation and progression [32, 33].

Figure III-3: The expression of NICD and Hes1 in MDA-MB-231 and SUM-159 cells by treatment with rhJag1.

[32]Shao S, Zhao X, Zhang X, et al. Notch1 signaling regulates the epithelial-mesenchymal transition and invasion of breast cancer in a Slug-dependent

manner. *Molecular cancer*, 2015, 14(1): 28.

[33] Yamamoto M, Taguchi Y, Ito-Kureha T, et al. NF- κ B non-cell-autonomously regulates cancer stem cell populations in the basal-like breast cancer subtype. *Nature communications*, 2013, 4: 2299.

(3) anti-Notch1cd antibody worked and was specific;

Answer: In our study, we used two anti-NICD (anti-Notch1cd) antibodies for immunoblotting, ChIP and immunohistochemical analysis. To confirm the specificity of the anti-NICD antibody (CST, #4147) for immunoblotting, MDA-MB-231 and SUM-159 cells were treated with γ -secretase inhibitor DAPT. The results showed that DAPT could block the cleavage of Notch1cd and thus downregulate Hes1 expression (Figure III-4), which is consistent with the previous studies that inhibition of Notch1 activity by DAPT downregulates NICD expression and inhibits tumorigenesis [34, 35].

Figure III-4: Immunoblotting for NICD (CST, #4147) and Hes1 in MDA-MB-231 and SUM-159 cells by treatment with DAPT.

To verify the specificity of the anti-NICD (CST, #4147) for ChIP, we used rabbit normal IgG and anti-p300 antibodies [36] as negative controls to perform the experiments. As shown in Figure III-5, the ChIP assay using anti-NICD (CST, #4147) demonstrated that NICD was specifically recruited to both Notch response elements (NREs) in the Zeb1 promoter, and this recruitment was further increased by rhJag1 treatment (Figure III-5). However, the results using rabbit normal IgG and anti-p300 antibodies were negative regardless of rhJag1 treatment, revealing that the anti-NICD antibody (CST, #4147) specifically works in the ChIP assay.

Figure III-5: ChIP assays for recruitment of NICD (CST, #4147) and p300 to the endogenous Zeb1

promoter in MDA-MB-231 cells by treatment with rhJag1. $**P < 0.01$, $***P < 0.001$ vs. the respective control by unpaired Student's *t*-test.

For the anti-NICD antibody (Abcam, ab8925) used in immunohistochemical analysis, the expression of NICD was located both in the nucleus and cytoplasm (Figure III-6). In particular, we observed a predominant expression of NICD in the nucleus in the samples with high NICD expression. This is consistent with the previous studies that, in addition to canonical nuclear signaling, Notch1 also drives non-canonical, cytoplasmic and mitochondrial signals in TNBC stem cells [37-41].

Figure III-6: Immunohistochemical analysis using anti-NICD (Abcam, ab8925) in breast cancer specimen. Scale bars, 50 μ m.

- [34] Astudillo L, Da Silva T G, Wang Z, et al. The small molecule IMR-1 inhibits the notch transcriptional activation complex to suppress tumorigenesis. *Cancer research*, 2016, 76(12): 3593-3603.
- [35] Liubomirski Y, Lerrer S, Meshel T, et al. Notch-mediated tumor-stroma-inflammation networks promote invasive properties and CXCL8 expression in triple-negative breast cancer. *Frontiers in immunology*, 2019, 10: 804.
- [36] Zhang X, Zhang Z, Zhang Q, et al. ZEB1 confers chemotherapeutic resistance to breast cancer by activating ATM. *Cell death & disease*, 2018, 9(2): 57.
- [37] Andersen P, Uosaki H, Shenje L T, et al. Non-canonical Notch signaling: emerging role and mechanism. *Trends in cell biology*, 2012, 22(5): 257-265.
- [38] Ayaz F, Osborne B A. Non-canonical notch signaling in cancer and immunity. *Frontiers in oncology*, 2014, 4: 345.
- [39] Liu Z, Teng L, Bailey S, et al. Epithelial transformation by KLF4 requires Notch1 but not canonical Notch1 signaling. *Cancer biology & therapy*, 2009, 8(19): 1840-1851.
- [40] Jin S, Mutvei A P, Chivukula I V, et al. Non-canonical Notch signaling activates IL-6/JAK/STAT signaling in breast tumor cells and is controlled by p53 and IKK α /IKK β . *Oncogene*, 2013, 32(41): 4892.
- [41] Hossain F, Sorrentino C, Ucar D A, et al. Notch Signaling Regulates

Mitochondrial Metabolism and NF- κ B activity in Triple-Negative Breast Cancer Cells via IKK α -dependent Non-Canonical Pathways. *Frontiers in oncology*, 2018, 8: 575.

(4) the co-culture system with HUVEC and MBA-MB-231 worked and was appropriate to evaluate their purpose;

Answer: To evaluate TC-EC interaction in the co-culture system, we performed the immunofluorescence staining for CD44, CD24 and mCherry using the co-culture of MDA-MB-231 with HUVEC-mCherry *in vitro*. The analysis of confocal microscopy demonstrated that the direct interaction with HUVEC-mCherry induced an increased CD44⁺CD24⁻ breast CSC phenotype in contact MDA-MB-231 but not in free MDA-MB-231 (Figure III-7a and b). Moreover, CD44⁺CD24⁻ breast CSCs and non-CSCs populations were further derived by cell sorting. The quantitative PCR and immunoblotting revealed elevated expression of Zeb1, NICD and VEGFA in breast CSCs compared with those in non-CSC populations (Figure III-7c and d).

Figure III-7: The juxtacrine interaction with HUVEC-mCherry induces breast CSC phenotype (a and b) and consequent upregulation of Zeb1, NICD, Hes1 and VEGFA (c and d) in MDA-MB-231 cells. $**P < 0.001$, $***P < 0.001$ vs. the respective control by unpaired Student's *t*-test. Scale bars, 20 μ m (a).

Based on this, our results also demonstrated that specific interference of Jag1, but not Dll4, in HUVEC-mCherry markedly abrogated the upregulation of Zeb1 (Figure III-8a and b). On the other hand, Zeb1 upregulation by cultured HUVEC-mCherry was specifically weakened in shNotch1-interfered MDA-MB-231 compared to those cells expressing shRNAs for Notch2 or Notch3 (Figure III-8c and d).

Figure III-8: Endothelial Jag1 confers upregulation of Zeb1 through activating the Notch1 signaling in MDA-MB-231 cells. $**P < 0.001$, $***P < 0.001$ vs. the respective control by unpaired Student's *t*-test.

Of note, MDA-MB-231 were also treated with conditioned medium (CM) derived from HUVECs expressing shRNAs for Jag1 or Dll4, whereas the alternations of Zeb1 expression were not evident (Figure III-9a and b). For the above experiments, we respectively used scramble shRNAs as control and two independent shRNAs for Notch1-3, Jag1 and Dll4 to confirm the results. We also conducted the coculture system of HUVECs with SUM-159 cells and obtained similar results (Figure III-9c and d).

Figure III-9: The expression of Zeb1 in MDA-MB-231 (a and b) and SUM-159 (c and d) cells treated with conditioned medium derived from HUVECs expressing shRNAs for Jag1 or DII4.

Collectively, these observations demonstrated that juxtacrine activation of Notch1 signaling by endothelial Jag1 induces Zeb1 expression in breast cancer cells, which in turn contributes to the establishment and maintenance of their CSC phenotypes. Moreover, our results are consistent with the previous studies using the co-culture system with HUVECs and MBA-MB-231 (or MCF-7) to demonstrate a crosstalk mechanism between tumor and microenvironment where tumor-stimulated mesenchymal modulation of endothelial cells enhances the constitution of a transient mesenchymal/endothelial niche leading to significant increase in breast cancer proliferation, stemness, and invasiveness [42].

[42] Ghiabi P, Jiang J, Pasquier J, et al. Breast cancer cells promote a notch-dependent mesenchymal phenotype in endothelial cells participating to a pro-tumoral niche. *Journal of translational medicine*, 2015, 13(1): 27.

(5) the co-transplantation experiment with HUVEC and MBA-MB-231 into mammary fat pad worked well.

Answer: Consequently, the above results were also confirmed in the co-injection xenograft model *in vivo*. The extreme limiting dilution assays supported our notion that endothelial Jag1-induced activation of Notch1 confers breast cancer initiation and progression in a Zeb1-dependent manner. Moreover, mice with xenograft breast cancers were further treated with anti-Notch1 and Avastin, alone or in combination, showing reduced tumor growth and prolonged survival rate. The immunoblotting analysis demonstrated strong downregulation of Zeb1 expression and decrease in Notch1 activity and angiogenesis especially in tumors from the combinational treatment group. Considering that treatment with anti-Notch1 and Avastin predominantly functions to counteract Jag1-Notch1-Zeb1-VEGFA-mediated angiocrine signaling in fostering tumor perivascular niches, these results collectively

confirmed that disrupting the interaction of HUVECs with MBA-MB-231 reduces the incidence of breast tumorigenesis in the co-injection xenograft model *in vivo*. In line, Ghiabi *et al.* reported that tumor-stimulated mesenchymal endothelial cells are capable of constituting a pro-tumoral niche responsible for increasing breast cancer cell proliferation, mammary stem cell self-renewal and pro-metastatic properties using this co-culture system *in vivo* [42].

In addition to their experimental problems, I very wonder whether cancer stem cells contact to endothelial cells. In this case, do cancer stem cells experience hypoxia?

Answer: To evaluate TC-EC interaction *in vivo*, we performed immunofluorescent staining in tumors from PyMT and PyMT;*Zeb1*^{CKO} mice using anti-CD31, anti-ALDH1 and anti-NICD antibodies. As shown in Figure III-10a and b, we observed that breast cancer cells expressing ALDH1 and NICD predominantly colocalized with CD31-positive endothelial cells in tumors from PyMT mice, showing that tumor-initiating cells with elevated ALDH1 and Notch1 activity are localized to adjacent perivascular niches which could directly interact with each other. However, this perivascular location of ALDH1 and NICD was significantly attenuated by *Zeb1* depletion in PyMT;*Zeb1*^{CKO} tumors. Our results also revealed strongly decreased microvessel density and reduced CSC phenotype in PyMT;*Zeb1*^{CKO} tumors, highlighting that depletion of *Zeb1* contributes to the impairment of tumor perivascular niche and the inhibition in CSC properties. In line, the immunofluorescence staining for CD31 and CD44 confirmed the colocalization of CD44⁺ breast CSCs in the perivascular region close to endothelial cells in PyMT tumors, whereas this contact between breast CSCs and endothelial cells was significantly disrupted in PyMT;*Zeb1*^{CKO} tumors (Figure III-10c). These observations are consistent with several previous studies showing that the perivascular niches are localized to adjacent tumor-initiating cells which could directly interact with each other in certain types of cancers [43-47]. For example, the vascular niche localized in the subventricular zone houses neural stem and progenitor cells that could initiate gliomas [43, 44].

Figure III-10: The juxtacrine interaction with perivascular niches confers Notch1-Zeb1 activation and breast CSC property *in vivo*. Immunofluorescence staining for CD31, ALDH1 (a), NICD (b) and CD44 (c) in breast cancer tissue (n = 5 PyMT, 5 PyMT;Zeb1^{cKO}). The highlighted regions in the left panels are enlarged in the right panels. * $P < 0.05$, ** $P < 0.01$, *** $P < 0.001$ vs. the respective control by unpaired Student's *t*-test. Scale bars, 20µm (a and c); 50µm (b).

It has been demonstrated that the tumor vasculature is characterized by dilated, tortuous and disorganized blood vessels. Vascular immaturity and lack of mural cell (*e.g.*, pericytes) association lead to excessive permeability, poor perfusion and increased hypoxia. Thus, the atypical morphological and functional features of vascular normalization disruption, including decreased vessel coverage, increased vessel leakage and enhanced hypoxia, have been identified in tumor angiogenesis [45]. These are consistent with our study to demonstrate the alterations of vascular coverage, leakiness and hypoxic condition in tumors from PyMT and PyMT;Zeb1^{cKO} mice. As shown in Figure III-11a and b, the results confirmed significantly increased perivascular coverage but decreased dextran leakiness in tumors from PyMT;Zeb1^{cKO} mice compared with those from PyMT mice. Moreover, PyMT;Zeb1^{cKO} tumors demonstrated a decreased hypoxic condition by pimonidazole (PIMO) staining compared with the PyMT tumors (Figure III-11c). These observations revealed that ectopic Zeb1 plays a predominant role in the production of an abnormal blood vessel

network with impaired permeability, microvascular perfusion and hypoxia in breast cancer [46].

Of note, our GSEA analysis also showed that *Zeb1* depletion was significantly associated with reduced HIF-1 α signature in PyMT;*Zeb1*^{CKO} cells compared with that in PyMT cells (Figure III-11d). In line, breast CSCs and non-CSCs populations were respectively derived from PyMT tumor cells. The expression of Zeb1 and HIF-1 α were confirmed to be upregulated in breast CSCs compared with those in non-CSC populations (Figure III-11e and f), implying that ectopic Zeb1 might contribute a hypoxic condition in breast CSCs via the potential downstream pathways, such as the HIF-1 α signaling axis.

Figure III-11: Zeb1 confers the vascular abnormalization with excessive permeability, poor perfusion and increased hypoxia in breast cancer. Vessel normalization (a), dextran leakage assay (b) and PIMO staining for hypoxic area (c) in breast cancer tissue (n = 5 PyMT, 5 PyMT;Zeb1^{ckO}). ***P* < 0.01, ****P* < 0.001 vs. the respective control by unpaired Student's *t*-test. Scale bars, 200μm (a); 50μm (b and c). (d) Enrichment of gene signature associated with HIF-1α-related pathway in PyMT;Zeb1^{ckO} vs. PyMT cells by GSEA analysis. (e and f) Expression of Zeb1 and HIF-1α in CD44⁺CD24⁻ breast CSCs or non-CSCs (n = 3 PyMT, 3 PyMT;Zeb1^{ckO}). **P* < 0.05, ***P* < 0.01, ****P* < 0.001 vs. the respective control by unpaired Student's *t*-test.

Additionally, we provided evidence that the microenvironmental abundance of VEGFA induced by ectopic Zeb1 promotes continual tumor angiogenesis. In line with this, the previous studies have shown that hypoxia is associated with enhanced breast CSC and metastatic phenotypes through dysregulation of VEGFA and tumor angiogenesis [47-52]. Thus, despite their localization adjacent to the perivascular niches, breast CSCs with ectopic Zeb1 experience the alterations related to vessel normalization disruption, including decreased vascular perfusion, enhanced hypoxia and increased vessel leakage. Collectively, we might propose an alternative mechanism of Zeb1-induced CSC properties through the juxtacrine interaction with the perivascular niches in breast cancer.

- [43] Calabrese C, Poppleton H, Kocak M, et al. A perivascular niche for brain tumor stem cells. *Cancer cell*, 2007, 11(1): 69-82.
- [44] Zhu T S, Costello M A, Talsma C E, et al. Endothelial cells create a stem cell niche in glioblastoma by providing NOTCH ligands that nurture self-renewal of cancer stem-like cells. *Cancer research*, 2011, 71(18): 6061-6072.
- [45] Viallard C, Larrivé B. Tumor angiogenesis and vascular normalization: alternative therapeutic targets. *Angiogenesis*, 2017, 20(4): 409-426.
- [46] Kirane A, Toombs J E, Larsen J E, et al. Epithelial–mesenchymal transition increases tumor sensitivity to COX-2 inhibition by apricoxib. *Carcinogenesis*, 2012, 33(9): 1639-1646.
- [47] Schwab L P, Peacock D L, Majumdar D, et al. Hypoxia-inducible factor 1 α promotes primary tumor growth and tumor-initiating cell activity in breast cancer. *Breast Cancer Research*, 2012, 14(1): R6.
- [48] Mimeault M, Batra S K. Hypoxia-inducing factors as master regulators of stemness properties and altered metabolism of cancer-and metastasis-initiating cells. *Journal of cellular and molecular medicine*, 2013, 17(1): 30-54.
- [49] Philip B, Ito K, Moreno-Sánchez R, et al. HIF expression and the role of hypoxic microenvironments within primary tumours as protective sites driving cancer stem cell renewal and metastatic progression. *Carcinogenesis*, 2013, 34(8): 1699-1707.
- [50] Semenza G L. Regulation of the breast cancer stem cell phenotype by hypoxia-inducible factors. *Clinical Science*, 2015, 129(12): 1037-1045.
- [51] da Motta L L, Ledaki I, Purshouse K, et al. The BET inhibitor JQ1 selectively impairs tumour response to hypoxia and downregulates CA9 and angiogenesis in triple negative breast cancer. *Oncogene*, 2017, 36(1): 122.
- [52] Liang H, Xiao J, Zhou Z, et al. Hypoxia induces miR-153 through the IRE1 α -XBP1 pathway to fine tune the HIF1 α /VEGFA axis in breast cancer

angiogenesis. *Oncogene*, 2018, 37(15): 1961.

Do cancer stem cells produce VEGF? Is there any evidence that Jag1 expression in endothelial cells is induced by VEGF that was produced by cancer stem cells?

Answer: Single-cell suspension were prepared using PyMT tumors. CD44⁺CD24⁻ breast CSCs and non-CSCs populations were derived by cell sorting. The mRNA and protein expression of Zeb1, NICD, Hes1 and VEGFA were significantly upregulated in breast CSCs compared with those in non-CSC populations (Figure III-12a and b). These results are consistent with those in breast CSCs from MDA-MB-231 (Figure III-12c and d) and SUM-159 cells (Figure III-12e and f), demonstrating that breast CSCs with dysregulated Notch1-Zeb1 activity predominantly produces VEGFA. To further examine that VEGFA contributes to endothelial Jag1 expression, HUVECs were treated with rhVEGFA. The result showed that Jag1 expression was upregulated upon rhVEGFA treatment (Figure III-12g and h), which is consistent with the previous report by Kirane *at al.* that VEGFA enhances Jag1 expression via a paracrine action in adventitial microvascular endothelial cells [46]. In line, the expression of Jag1 was also increased in the presence of conditioned medium derived from MDA-MB-231 cells; however, this effect was weakened by Avastin addition (Figure III-12g and h). We also performed these experiments in SUM-159 cells and obtained similar results (Figure III-12i and j). On the other hand, CD31⁺ endothelial cells were isolated from PyMT;*Zeb1*^{CKO} and PyMT tumors. The results demonstrated that the fraction of CD31⁺ endothelial cells and the accompanied Jag1 expression were significantly reduced by *Zeb1* depletion in PyMT;*Zeb1*^{CKO} tumors (Figure III-12k and l). Taken together with our immunofluorescent analysis showing that breast CSCs are localized to adjacent endothelial cells via a juxtacrine interaction *in vitro* and *in vivo*, we collectively demonstrated that breast CSCs with dysregulated Notch1-Zeb1 activity predominantly produces VEGFA and thus induces endothelial Jag1 expression in the adjacent perivascular niches.

Figure III-12: VEGFA-derived from breast CSCs confers the upregulation of endothelial Jag1. (a and b) The expression of Hes1, NICD, Zeb1 and VEGFA in CD44⁺CD24⁻ breast CSCs or non-CSCs (n = 5 PyMT, 5 PyMT;*Zeb1*^{CKO}). (c and d) The expression of Hes1, NICD, Zeb1 and VEGFA in CD44⁺CD24⁻ breast CSCs or non-CSCs from MDA-MB-231 cells. (e and f) The expression of Hes1, NICD, Zeb1 and VEGFA in CD44⁺CD24⁻ breast CSCs or non-CSCs from SUM-159 cells. (g and h) The expression of Jag1 in HUVECs that were treated with rhVEGF or conditioned medium derived from MDA-MB-231 in the presence or absence of Avastin. (i and j) The expression of Jag1 in HUVECs that were treated with rhVEGF or conditioned medium derived from SUM-159 in the presence or absence of Avastin. (k) Flow cytometry analysis of the CD31⁺ endothelial cell fraction (n = 5 PyMT, 5 PyMT;*Zeb1*^{CKO}). (l) The expression of Jag1 in CD31⁺ endothelial cells (n = 5 PyMT, 5 PyMT;*Zeb1*^{CKO}). **P < 0.01, ***P < 0.001 vs. the

respective control by unpaired Student's *t*-test.

In immunohistochemical analysis of human breast cancer tissues, Notch1icd is expressed in the nucleus or cytoplasm of cancer stem cells? Are Notch1icd-expressing cells cancer stem cells? If so, I very wonder the frequency of cancer stem cells in the cancer tissues? Was anti-Notch1icd antibody specific? There are too many technical, experimental, and conceptual problems.

Answer: Based on our results of immunohistochemical staining, the expression of NICD (anti-Notch1icd, Abcam, ab8925) was located both in the nucleus and cytoplasm. In particular, we observed a predominant expression of NICD in the nucleus in the samples with high NICD expression. This is consistent with the previous studies that, in addition to canonical nuclear signaling, Notch1 also drives non-canonical, cytoplasmic and mitochondrial signals in TNBC stem cells [53-55]. Furthermore, we demonstrated a positive correlation between the expression of NICD, Zeb1 and ALDH1 in 175 cases of primary breast carcinoma, indicating strongly elevated expression of NICD and Zeb1 in poorly differentiated high-grade tumors with high ALDH1 activity. Taken together with our results showing a strong correlation between high expression of NICD and Zeb1 with the TNBC subtype, we proposed that dysregulated Notch1-Zeb1 activity is functionally linked to tumor stem cell traits in breast cancer. Indeed, a variety of study have revealed that high-grade tumors and TNBCs are often enriched with abundant CSC populations and are correlated with increased tumor angiogenesis [56-58]. Moreover, this anti-NICD antibody has been applied to indicate the Notch1 activity of CSCs in various cancer types including breast cancer [59-61], showing a similar pattern of IHC staining.

CSCs are a subpopulation of cancer cells that have acquired the stemness properties, including self-renewal, as well as the ability to generate more differentiated cells forming the bulk of the tumor. Although CSCs account for a small proportion, at the apex of the hierarchy, considered as “seeds” in the tumor microenvironment. Despite that CSC markers have proven useful for the enrichment of CSC populations, their utility is limited by variability of expression and regulation by environmental factors. So far, it is still impossible to make a definitive identification. Cancer cells, displaying protein markers including CD44 and ALDH, are able to generate tumors when transplanted into immunocompromised mice, and these tumors recapitulate the phenotypic heterogeneity of the initial tumor. Strictly speaking, tumor stem cells are included in these CD44- or ALDH-positive cell subsets, but these marker-positive cancer cells, same as NICD-positive cells, are not all CSCs.

[53]Jin S, Mutvei A P, Chivukula I V, et al. Non-canonical Notch signaling activates

- IL-6/JAK/STAT signaling in breast tumor cells and is controlled by p53 and IKK α /IKK β . *Oncogene*, 2013, 32(41): 4892.
- [54] Ayaz F, Osborne B A. Non-canonical notch signaling in cancer and immunity. *Frontiers in oncology*, 2014, 4: 345.
- [55] Hossain F, Sorrentino C, Ucar D A, et al. Notch Signaling Regulates Mitochondrial Metabolism and NF- κ B activity in Triple-Negative Breast Cancer Cells via IKK α -dependent Non-Canonical Pathways. *Frontiers in oncology*, 2018, 8: 575.
- [56] Chao C H, Chang C C, Wu M J, et al. MicroRNA-205 signaling regulates mammary stem cell fate and tumorigenesis. *The Journal of clinical investigation*, 2014, 124(7): 3093-3106.
- [57] Zhang P, Wei Y, Wang L, et al. ATM-mediated stabilization of ZEB1 promotes DNA damage response and radioresistance through CHK1. *Nature cell biology*, 2014, 16(9): 864.
- [58] Langer E M, Kendsersky N D, Daniel C J, et al. ZEB1-repressed microRNAs inhibit autocrine signaling that promotes vascular mimicry of breast cancer cells. *Oncogene*, 2018, 37(8): 1005.
- [59] Upadhyay P, Nair S, Kaur E, et al. Notch pathway activation is essential for maintenance of stem-like cells in early tongue cancer. *Oncotarget*, 2016, 7(31): 50437.
- [60] Zhong Y, Shen S, Zhou Y, et al. NOTCH1 is a poor prognostic factor for breast cancer and is associated with breast cancer stem cells. *OncoTargets and therapy*, 2016, 9: 6865.
- [61] Bayin N S, Frenster J D, Sen R, et al. Notch signaling regulates metabolic heterogeneity in glioblastoma stem cells. *Oncotarget*, 2017, 8(39): 64932.

Reviewers' comments:

Reviewer #1 (Remarks to the Author):

The revised manuscript addresses all my concerns.

Reviewer #1 (Feedback regarding Reviewer's #3 comments):

We are providing here some notes about Reviewer's #1 confidential communication with us regarding Reviewer's #3 comments, which you might find useful in case you decide to submit a revision of your manuscript. Regarding the specificity of the antibody you used to detect Notch1 activation, NICD, Reviewer#1 mentioned that there are two specific antibodies to detect NICD by IF. They are monoclonal antibodies from Cell Signaling and Abcam for the 4-amino acid epitope generated by gamma secretase cleavage (VLLS). We noticed those are different from the antibody you used in your work so Reviewer #1 suggests to perform some experiments to demonstrate the specificity of your stainings by, for example, using shRNA Notch1 cells compared to shControl cells. Reviewer #1 was not concerned about Notch activation in non-stem cells since, from her/his comments, MDA-MB231 possess endogenous activation of Notch1 and actually express Jagged1. Moreover, Reviewer #1 explained that MDA-MB231 cells contain a high fraction of stem-like cells. Regarding the methods used to define stem cells, Reviewer #1 supports the use of CD44^{high}/CD24^{low} markers but suggests additional experiments which would strength those observations such as using alternative stemness markers (CD90) or assessing mammosphere forming cells by limiting dilution, after co-culturing MDA-231 cells with shControl or shJagged1 HUVEC cells.

Reviewer #2 (Remarks to the Author):

The authors have satisfactorily addressed my concerns.

Reviewer #3 (Remarks to the Author):

Although the authors added tremendous efforts on this manuscript, the responses to most comments were not satisfied. For examples, In response to comment 3. Does the cultures of MDA-MB-231 or SUM-159 alone drive the cleavage of Notch1? How does it happen? Is 20nM DAPT enough to inhibit notch activation? Is there any evidence that NICD is the real Notch1cd? It needs to show full blot. For the ChIP, there is no match between panel a and panel b. There is no difference between control and rhJag1 in panel a, but huge difference in panel b. In the immunohistochemical analysis, there is no evidence that the signal is specific to Notch1cd.

In response to comment 4. What is the criteria and definition of CSC phenotype? How to discriminate non-CSC and CSC? How to know an increased CD44+CD24- breast CSC phenotype in contact MDA-MB-231 but not in free MDA-MB-231? How's NICD generated in non-CSC? How's Zeb1 expressed in non-CSC? It is very curious how NICD and Zeb1 are generated in non-CSC or MDA-MB-231 alone culture, which is contradict to their conclusion that juxtacrine activation of Notch1 signaling by endothelial Jag1 induces Zeb1 expression in breast cancer cells.

In response to comment 5. There is no evidence that the co-transplantation experiment with HUVEC and MBA-MB-231 into mammary fat pad worked well. Since there is no evidence that anti-NICD antibody works specifically, co-staining results with anti-NICD and anti-CD31 or anti-ALDH1 will not give any valuable conclusion.

Beside those comments, the responses to the rest of my comments was not satisfied.

Reviewer's comments (Reviewer #3):

In response to comment 3:

Does the cultures of MDA-MB-231 or SUM-159 alone drive the cleavage of Notch1?
How does it happen?

Answer: As shown in Figure I-1, MDA-MB-231 cells were cultured at different densities of $\leq 10\%$ (cell-free condition: almost no cell-contact) and $\geq 80\%$ (cell-contact condition), respectively. The results of immunoblotting showed that the expression level of NICD (Notch1icd) was upregulated under the cell-contact condition compared with that under the cell-free condition (Figure I-1b). Consequently, the expression of endogenous Notch1 ligands Jag1 and Dll4 were examined by quantitative PCR and immunoblotting in MDA-MB-231 cells. The results demonstrated that Jag1 expression was relatively higher in MDA-MB-231 (Figure I-1c and d). This is consistent with the previous reports showing that breast cancer cells including MDA-MB-231 possess endogenous activation of Notch1 and actually express Jag1 [1]. In addition, the expression of Jag1 was specifically knocked down in MDA-MB-231 (shJag1/231), followed by culturing at different densities (Figure I-1a and b). We found that the upregulation of NICD was attenuated in shJag1/231 under the cell-contact condition. We also performed these experiments in SUM-159 cells and obtained the same results (Figure I-2).

Figure I-1: Jag1 interference inhibits the activation of Notch1 signaling in MDA-MB-231 breast cancer cells. (a) The knockdown efficiency of specific shRNAs for Jag1 in MDA-MB-231. (b) The expression of Notch1, NICD (Notch1icd) and Hes1 in shCtrl/231 and shJag1/231 cells cultured at different densities. (c and d) The endogenous expression of Jag1 and Dll4 in MDA-MB-231 and SUM-159 cells by (c) quantitative PCR and (d) immunoblotting. Scale bars, 100 μ m.

Figure I-2: Jag1 interference inhibits the activation of Notch1 signaling in SUM-159 breast cancer cells. (a) The knockdown efficiency of specific shRNAs for Jag1 in SUM-159. (b) The expression of Notch1, NICD (Notch1icd) and Hes1 in shCtrl/159 and shJag1/159 cells cultured at different densities. (c) The endogenous expression of Jag1 and Dll4 in SUM-159 by quantitative PCR. Scale bars, 100 μ m.

- [1] Shao S, Zhao X, Zhang X, *et al.* Notch1 signaling regulates the epithelial-mesenchymal transition and invasion of breast cancer in a Slug-dependent manner. *Mol Cancer*. 2015; 14: 28.

Is 20nM DAPT enough to inhibit notch activation? Is there any evidence that NICD is the real Notch1icd? It needs to show full blot.

Answer: We apologize that we made a typo for the concentrations of DAPT. In the previous revision, we used 20-60 μ M DAPT to treat MDA-MB-231 cells, followed by detection of NICD expression using immunoblotting. In Figure I-3a, we showed the original records of our data. Moreover, we repeated the experiments with lower concentrations of DAPT to verify the inhibition in Notch1 activation. As shown in Figure I-3b, different concentrations (0, 5, 10 and 20 μ M) of DAPT were used to treat MDA-MB-231 in the presence or absence of rhJag1. The results of immunoblotting demonstrated that DAPT at the concentration of 20 μ M could efficiently block the cleavage of NICD (Notch1icd) and thus downregulate the expression of Hes1, which is a downstream target gene of the Notch1 signaling, regardless of rhJag1 addition (Figure I-3b). We also performed the experiments in SUM-159 cells and obtained the same results (Figure I-3c). The full blots for both cell lines were shown below. Our data are consistent with the previous studies showing that the inhibition of Notch1 activity by DAPT at the concentrations of 10 μ M downregulates Notch1icd expression and inhibits tumorigenesis in breast cancer [2].

In our study, the cleaved Notch1 rabbit mAb of CST #4147 was used for

immunoblotting assays to detect the activation of Notch1 in breast cancer cells. This is a specific antibody for Notch1icd. According to the datasheet, this antibody could detect the endogenous level of Notch1icd only when the human Notch1 is cleaved and released at Gly1753/Val1754 (the site of Gly1743/Val1744 for mouse Notch1). This antibody does not recognize the full-length Notch1 or its cuts at other sites. Several studies have used this antibody to examine the Notch1 activation in lung, bone and colon cancer cells and obtained the specific results [3-5]. Moreover, to confirm the specificity of this antibody in our immunoblotting assays, specific shRNAs for Notch1, Notch2 and Notch3 were introduced in MDA-MB-231 cells, followed by treatment with rhJag1. The results showed that specific knockdown of Notch1 significantly blocked the upregulation of Notch1icd in response to rhJag1 addition, whereas shNotch2 or shNotch3 interference was not as evident (Figure I-3d). We also performed these experiments in SUM-159 cells and obtained the same results (Figure I-3e).

Figure I-3: Notch1 interference inhibits the expression of NICD (Notch1icd) by treatment with rhJag1 in MDA-MB-231 and SUM-159 breast cancer cells. (a) The original records of the concentrations of DAPT. (b and c) The expression of NICD (Notch1icd) and Hes1 in (b)

MDA-MB-231 and (c) SUM-159 cells by treatment with different concentrations of rhJag1. (d) The expression of Notch1 and NICD (Notch1icd) in Notch1-, Notch2- or Notch3-interfered MDA-MB-231 cells by treatment with rhJag1. (e) The expression of Notch1 and NICD (Notch1icd) in Notch1-, Notch2- or Notch3-interfered SUM-159 cells by treatment with rhJag1.

- [2] Shin S, Kim K, Kim H, *et al.* Deubiquitylation and stabilization of Notch1 intracellular domain by ubiquitin-specific protease 8 enhance tumorigenesis in breast cancer. *Cell Death Differ.* 2020; 27: 1341-1354.
- [3] Stupnikov M, Yang Y, Mori M, *et al.* Jagged and Delta-like ligands control distinct events during airway progenitor cell differentiation. *Elife.* 2019; 8: e50487.
- [4] Jackstadt R, von Hooff S, Leach J, *et al.* Epithelial NOTCH Signaling Rewires the Tumor Microenvironment of Colorectal Cancer to Drive Poor-Prognosis Subtypes and Metastasis. *Cancer Cell.* 2019; 36, 319-336.
- [5] Fu R, Lv W, Xu Y, *et al.* Endothelial ZEB1 promotes angiogenesis-dependent bone formation and reverses osteoporosis. *Nat Commun.* 2020; 11: 460.

For the ChIP, there is no match between panel a and panel b. There is no difference between control and rhJag1 in panel a, but huge difference in panel b.

Answer: We repeated the experiment and reperformed the ChIP assay using the antibodies for NICD (CST #4147) and CBF1 in MDA-MB-231 cells in the presence or absence of rhJag1. The rabbit normal IgG was used as a negative control. As shown in Figure I-4, the results demonstrated that NICD and CBF1 was specifically recruited to both Notch response elements (NREs) in the Zeb1 promoter, and this recruitment was further increased by rhJag1 treatment. However, the results using rabbit normal IgG were negative regardless of rhJag1 treatment, revealing that the anti-NICD antibody specifically works in the ChIP assay. Taken together with our results of the promoter activity analysis of Zeb1, these observations collectively suggested that Notch1 activation by rhJag1 stimulates Zeb1 transcription by forming a NICD/CBF1 complex on the Zeb1 promoter.

Figure I-4: ChIP assays for the recruitment of NICD and CBF1 to the endogenous Zeb1 promoter in MDA-MB-231 cells by treatment with rhJag1. *** $P < 0.001$ vs. the respective control by unpaired Student's t -test.

In the immunohistochemical analysis, there is no evidence that the signal is specific to Notch1icd.

Answer: We used activated Notch1 rabbit mAb of Abcam ab8925 for the immunochemical analysis in breast cancer tissues. In the datasheet, it is noticed that the epitope (VLLSRKRRRQHGQC) is only exposed after γ -secretase cleavage and is not accessible in the uncleaved form of Notch1. We also searched but did not find that this peptide showed any homology with the sequences of Notch2 and Notch3 by NCBI_Blast (RID CBFSHXMD01R). Previous studies have shown that this antibody is specific for identifying activated Notch1 in oral epithelial cells [6]. Moreover, to verify its specificity for the immunohistochemistry analysis in our study, specific shRNAs for Notch1, Notch2 and Notch3 were introduced in MDA-MB-231 cells. The results of immunohistochemistry showed that the specific knockdown of Notch1 significantly reduced the expression of NICD (Notch1icd), whereas shNotch2 or shNotch3 interference was not as evident (Figure I-5a). We also performed these experiments in SUM-159 cells and obtained the same results (Figure I-5b).

Figure I-5: Immunohistochemical analysis using the anti-NICD antibody (Abcam, ab8925). (a) The immunohistochemical staining for NICD (Notch1icd) in Notch1-, Notch2- or Notch3-interfered MDA-MB-231 cells. (b) The immunohistochemical staining for NICD (Notch1icd) in Notch1-, Notch2- or Notch3-interfered SUM-159 cells. Scale bars, 20µm.

[6] Al-Attar A, Alimova Y, Kirakodu S, *et al.* Activation of Notch-1 in oral epithelial cells by *P. gingivalis* triggers the expression of the antimicrobial protein PLA2-IIA. *Mucosal Immunol.* 2018; 11: 1047-1059.

In response to comment 4:

What is the criteria and definition of CSC phenotype? How to discriminate non-CSC and CSC? How to know an increased CD44⁺CD24⁻ breast CSC phenotype in contact MDA-MB-231 but not in free MDA-MB-231?

Answer: CSCs are a subpopulation of cancer cells that have acquired the stemness properties, including self-renewal, as well as the ability to generate more differentiated cells forming the bulk of the tumor. Although CSCs account for a small proportion, at the apex of the hierarchy, considered as “seeds” in the tumor microenvironment [7]. Certain CSC markers, including the sphere-forming capacity, the CD44⁺CD24⁻ breast CSC population and the expression of stemness-related genes, have proven useful for the enrichment of CSC populations [8,9]. In our study, to discriminate CSCs and non-CSCs in the coculture system, the sub-populations of CD44⁺CD24⁻ and non-CD44⁺CD24⁻ MDA-MB-231 cells were derived by cell sorting respectively (Figure I-6a) [10]. The analysis of established breast CSC markers and properties revealed strong increases in the tumorsphere-forming capacity (Figure I-6b) and the expression of stemness-related genes (Figure I-6c) in CD44⁺CD24⁻ cells compared to those in non-CD44⁺CD24⁻ cells. Considering that MDA-MB-231 cells are TNBCs and contain a high fraction of stem-like cells [11], these results demonstrated that breast CSCs are predominantly enriched in the CD44⁺CD24⁻ population. We also performed these experiments in SUM-159 cells and obtained the same results (Figure I-6d).

Figure I-6: Identification of breast CSC properties in the coculture system. (a) The sub-populations of non-CD44⁺CD24⁻ (non-CSC population) and CD44⁺CD24⁻ (CSC population) MDA-MB-231 cells derived by cell sorting in the coculture system. (b) The tumorsphere-forming capacity by CSC frequency analysis in non-CSCs and CSCs derived from MDA-MB-231 cells. (c) The expression of Sox2, Oct4 and Nanog by quantitative PCR and immunoblotting in non-CSCs and CSCs derived from MDA-MB-231 cells. (d) The expression of Sox2, Oct4 and Nanog by quantitative PCR and immunoblotting in non-CSCs and CSCs derived from SUM-159 cells. ** $P < 0.01$, *** $P < 0.001$ vs. the respective control by unpaired Student's t -test.

In addition, we found that coculture with HUVECs led to increased CD44⁺CD24⁻ breast CSC population in MDA-MB-231 cells by flow cytometry analysis; however, the effect of conditioned medium derived from HUVECs was not as evident (Figure I-7a and b). Importantly, to further evaluate TC-EC juxtacrine interaction and its contribution to breast CSC property in the coculture system, we performed the immunofluorescence staining for CD44, CD24 and mCherry using the coculture of MDA-MB-231 with HUVEC-mCherry *in vitro*. The analysis of confocal microscopy demonstrated that the direct interaction with HUVEC-mCherry induced an increased CD44⁺CD24⁻ cell population up to approximately 26% in contact MDA-MB-231; however, the percentage of CD44⁺CD24⁻ population was about 8% in free MDA-MB-231 (Figure I-7c and d). Moreover, CD44⁺CD24⁻ breast CSCs and non-CSCs populations were derived by cell sorting. The quantitative PCR and immunoblotting revealed elevated expression of Zeb1, NICD and VEGFA in breast CSCs compared with those in non-CSC populations (Figure I-7e), which is consistent with our notion that activation of Notch1 by TC-EC juxtacrine interaction confers breast CSC phenotypes and VEGFA production in a Zeb1-dependent manner. We also performed these experiments in SUM-159 cells and obtained the same results (Figure I-7f).

Figure I-7: The juxtacrine interaction with HUVECs induces breast CSC properties in the coculture system. (a and b) The percentage of CD44⁺CD24⁻ MDA-MB-231 sub-population by flow cytometry analysis. (c and d) The juxtacrine interaction with HUVECs induces CD44⁺CD24⁻ CSC phenotype in MDA-MB-231 cells. Red arrow: HUVEC-mCherry; yellow arrow: CD44⁺CD24⁻ MDA-MB-231 cancer stem-like cells. Scale bars, 20 μ m. (e and f) The expression of NICD (Notch1icd), Zeb1 and VEGFA by quantitative PCR and immunoblotting in (e) MDA-MB-231 and (f) SUM-159 cells cocultured with HUVECs. ** $P < 0.001$, *** $P < 0.001$ vs. the respective control by unpaired Student's *t*-test.

In line, we co-cultivated shJag1/HUVEC-mCherry or the respective control HUVEC-mCherry with MDA-MB-231 under anchorage-independent conditions to expand mammary stem cells as “mammo-angiospheres” (Figure I-8a). The results showed that shJag1/HUVECs significantly attenuated “mammo-angiosphere” formation of MDA-MB-231 compared to shCtrl/HUVECs (Figure I-8b). These observations indicated that TC-EC crosstalk potentially modulates breast CSC phenotype; contact with endothelial cells triggers mammosphere formation and self-renewal capacity in MDA-MB-231 cells. Thus, contact with endothelial cells seems to be crucial for the initiation and maintenance of breast CSC properties and may be regarded as an approach for treating cancer.

Figure I-8: Endothelial cells provide a pro-tumoral niche for mammo-angiosphere formation of MDA-MB-231 cells in the coculture system. (a) Schematic representation of the experimental procedures. (b and c) Mammosphere analysis of MDA-MB-231 cells cocultured with shCtrl/HUVECs or shJag1/HUVECs. Red arrow: mammo-angiospheres; green arrow: mammospheres. Scale bars, 100 μ m. *** $P < 0.001$ vs. the respective control by unpaired Student's *t*-test.

Of note, we also used CD90 as a breast CSC marker to verify the results [12]. As shown in Figure I-9a and b, coculture with HUVECs led to increased percentage of CD90⁺ population in MDA-MB-231 cells, whereas this effect was remarkably abolished by addition of anti-Notch1 neutralizing antibody. Similar results were also observed in SUM-159 cells (Figure I-9c and d), demonstrating that

endothelial-induced juxtacrine activation of Notch1 confers breast CSC phenotypes. Moreover, we observed strong decreases in the CD90⁺ breast CSC population in tumor cells from PyMT;*Zeb1*^{CKO} mice compared to those from PyMT mice (Figure I-9e and f), supporting the importance of *Zeb1* in the acquisition of CSC properties by breast cancer cells, which leads to tumor initiation and progression.

Figure I-9: The verification of CD90 as a CSC marker in the coculture system *in vitro* and in PyMT and PyMT;*Zeb1*^{CKO} mice *in vivo*. (a and b) The percentage of CD90⁺ MDA-MB-231

sub-population by flow cytometry analysis. (c and d) The percentage of CD90⁺ SUM-159 sub-population by flow cytometry analysis. (e and f) The percentage of CD90⁺ PyMT and PyMT;*Zeb1*^{CKO} sub-population by flow cytometry analysis. ***P* < 0.01, ****P* < 0.001 vs. the respective control by unpaired Student's *t*-test.

- [7] Reya T, Morrison S, Clarke M, *et al.* Stem cells, cancer, and cancer stem cells. *Nature*. 2001;414: 105-111.
- [8] Scheel C, Eaton EN, Li S, *et al.* Paracrine and autocrine signals induce and maintain mesenchymal and stem cell states in the breast. *Cell*. 2011; 145: 926-940.
- [9] Pattabiraman D, Weinberg R. Tackling the cancer stem cells - what challenges do they pose? *Nat Rev Drug Discov*. 2014; 13: 497-512.
- [10] Zheng F, Yue C, Li G, *et al.* Nuclear AURKA acquires kinase-independent transactivating function to enhance breast cancer stem cell phenotype. *Nature Commun*. 2016; 7: 1-17.
- [11] Bieri B, Pierce SE, Kroeger C, *et al.* Integrin-β4 identifies cancer stem cell-enriched populations of partially mesenchymal carcinoma cells. *Proc Natl Acad Sci*. 2017; 114: E2337-E2346.
- [12] Lu H, Clauser K, Tam W, *et al.* A breast cancer stem cell niche supported by juxtacrine signalling from monocytes and macrophages. *Nat Cell Biol*. 2014; 16: 1105-1117.

How's NICD generated in non-CSC? How's *Zeb1* expressed in non-CSC? It is very curious how NICD and *Zeb1* are generated in non-CSC or MDA-MB-231 alone culture, which is contradict to their conclusion that juxtacrine activation of Notch1 signaling by endothelial *Jag1* induces *Zeb1* expression in breast cancer cells.

Answer: Previous studies have demonstrated that MDA-MB-231 cells endogenously possess Notch1 activation and actually express *Jag1* [1]. In line, we detected the expression of NICD (Notch1cd) in parental MDA-MB-231 as well as in CSC (CD44⁺CD24⁻) and non-CSC (non-CD44⁺CD24⁻) sub-populations derived from MDA-MB-231. The results of quantitative PCR and immunoblotting demonstrated that CD44⁺CD24⁻ breast CSCs predominantly expressed a higher level of NICD and *Zeb1* compared with those in parental MDA-MB-231 and non-CSC sub-population (Figure I-10a). We also performed these experiments in SUM-159 cells and obtained the same results (Figure I-10b). This is consistent with the notion that MDA-MB-231 cells contain a high fraction of stem-like cells with elevated *Zeb1* expression [13]. Considering that heterogeneity is a hallmark of cancer even in the sub-populations of CSCs and/or non-CSCs, these observations might support a dynamic model in which the interconversions between CSC and non-CSC states rely on the heterogenic Notch1/*Zeb1* signaling.

Figure I-10: The expression of NICD (Notch1icd) and Zeb1 in parental, non-CSCs (non-CD44⁺CD24⁻ cells) and CSCs (CD44⁺CD24⁻ cells) derived from (a) MDA-MB-231 and (b) SUM-159 cells. ** $P < 0.01$, *** $P < 0.001$ vs. the respective control by unpaired Student's t -test.

As previously stated by Weinberg, *et al.* [14], in certain subtypes of breast cancer (e.g. the basal-like cell lines MDA-MB-231 and SUM-159), Zeb1 is a major determinant of cancer cell plasticity. They found that plastic non-CSCs maintain the Zeb1 promoter in a bivalent/poised chromatin configuration, enabling them to respond readily to microenvironmental signals, such as TGF- β . In response, the Zeb1 promoter converts from a bivalent/poised to active chromatin configuration, Zeb1 transcription increases, and non-CSCs (CD44^{low}) subsequently enter the CSC (CD44^{high}) state. In this regard, it is plausible that basal non-CSCs located in the TME (e.g. perivascular niches) that is rich in EMT-inducing heterotypic signals may respond to local stimuli by switching to a CSC state; the resulting cells may then significantly enhance the aggressiveness of the tumors in which they reside. These also support our notion that endothelial-derived activation of Notch1/Zeb1 signaling contributes to breast CSC phenotypes, which could also be a dynamic state in the perivascular niches of breast cancer.

[13]Trapp EK, Majunke L, Zill B, *et al.* LKB1 pro-oncogenic activity triggers cell survival in circulating tumor cells. *Mol Oncol.* 2017; 11: 1508-1526.

[14]Chaffer C, Marjanovic N, Lee T, *et al.* Poised chromatin at the ZEB1 promoter enables breast cancer cell plasticity and enhances tumorigenicity. *Cell.* 2013; 154: 61-74.

In response to comment 5:

There is no evidence that the co-transplantation experiment with HUVEC and MBA-MB-231 into mammary fat pad worked well.

Answer: To investigate the role of endothelial cells in creating a pro-tumoral niche *in vivo*, we injected MDA-MB-231 to the mammary fat pad of NOD/SCID mice with or without HUVEC-mCherry cells. The mice that were co-injected with MDA-MB-231 and shCtrl/HUVECs showed significantly higher tumor burden with tumors weighing 2.8-fold higher than those mice that were only inoculated with MDA-MB-231 cells, while strong inhibition of tumor growth was achieved by co-injection with shJag1/HUVECs (Figure I-11a and b). Consistently, analysis of tumor volume displayed similar results showing that Jag1 depletion in HUVECs strongly attenuated the pro-tumoral effect in the co-injection xenograft tumors (Figure I-11c). Furthermore, the tumors sections were stained with mCherry and CD44 to show the existence of viable capillaries and the stemness properties in xenograft tumors. The results of immunofluorescent analysis demonstrated that co-injection with shCtrl/HUVECs generated functional and viable vessels in xenograft tumors (Figure I-11d). Importantly, our results also confirmed the colocalization of CD44⁺ cancer stem-like cells in the perivascular region close to HUVEC-mCherry in tumors co-injected with MDA-MB-231 and shCtrl/HUVECs; however, this contact between breast CSCs and endothelial cells was significantly disrupted in tumors co-injected with MDA-MB-231 and shJag1/HUVECs.

Figure I-11: Endothelial-derived Jag1 confers the pro-tumoral niche formation and breast tumor growth *in vivo*. (a and b) Tumor volume of BALB/c mice co-injected with MDA-MB-231 and shCtrl/HUVECs or shJag1/HUVECs; n = 5. (c) Tumor weight of BALB/c mice co-injected with MDA-MB-231 and HUVECs; n = 5. (d) Immunofluorescence staining for mCherry and CD44 in tumor sections; n = 5. Scale bars, 20µm. (e) The percentage of CD44⁺ tumor cell area in tumor sections. **P* < 0.05, ***P* < 0.01, ****P* < 0.001 vs. the respective control by unpaired Student's *t*-test.

Based on these, our results of the extreme limiting dilution assays further supported the notion that endothelial Jag1-induced activation of Notch1 confers breast cancer stemness and initiation in a Zeb1-dependent manner. Moreover, mice with xenograft breast cancers were treated with anti-Notch1 and Avastin, alone or in combination, showing reduced tumor growth and prolonged survival rate. The immunoblotting analysis revealed strong downregulation of Zeb1 expression and decreases in Notch1 activity and angiogenesis especially in tumors from the combinational treatment group. Considering that treatment with anti-Notch1 and Avastin predominantly functions to counteract Jag1-Notch1-Zeb1-VEGFA-mediated angiocrine signaling in fostering tumor perivascular niches, these data collectively confirmed that disrupting the interaction of HUVECs with MBA-MB-231 reduces the incidence of breast tumorigenesis in the co-injection xenograft model *in vivo*. In line, Ghiabi *et al.* reported that tumor-stimulated mesenchymal endothelial cells are capable of constituting a pro-tumoral niche responsible for increasing breast cancer cell proliferation, mammary stem cell self-renewal and pro-metastatic properties using the same co-culture system *in vivo* [15].

[15]Ghiabi P, Jiang J, Pasquier J, *et al.* Breast cancer cells promote a notch-dependent mesenchymal phenotype in endothelial cells participating to a pro-tumoral niche. *J Transl Med.* 2015; 13: 27.

Since there is no evidence that anti-NICD antibody works specifically, co-staining

results with anti-NICD and anti-CD31 or anti-ALDH1 will not give any valuable conclusion.

Answer: The specificity of anti-NICD antibodies of CST #4147 and Abcam ab8925 have been verified. The results are shown in Figures I-3 and I-5.

Beside those comments, the responses to the rest of my comments was not satisfied.

Answer: We feel that there might be a typo in this comment. However, if the reviewer is truly unsatisfied with the rest of our responses, we will be happy to address them if the reviewer provides the details.

REVIEWERS' COMMENTS:

Reviewer #3 (Remarks to the Author):

I'm satisfied now.

Responses to reviewers:

Reviewer #3:

I'm satisfied now.

Answer: We appreciate your positive review.